# LD-EnSF: Synergizing Latent Dynamics with Ensemble Score Filters for Fast Data Assimilation with Sparse Observations

**Pengpeng Xiao**[1,2]    **Phillip Si**[1]    **Peng Chen**[1]*
[1]Georgia Institute of Technology    [2]Yale University
`p.xiao@yale.edu,{psi6,pchen402}@gatech.edu`

## Abstract

Data assimilation techniques are crucial for accurately tracking complex dynamical systems by integrating observational data with numerical forecasts. Recently, score-based data assimilation methods emerged as powerful tools for high-dimensional and nonlinear data assimilation. However, these methods still incur substantial computational costs due to the need for expensive forward simulations. In this work, we propose LD-EnSF, a novel score-based data assimilation method that eliminates the need for full-space simulations by evolving dynamics directly in a compact latent space. Our method incorporates improved Latent Dynamics Networks (LDNets) to learn accurate surrogate dynamics and introduces a history-aware LSTM encoder to effectively process sparse and irregular observations. By operating entirely in the latent space, LD-EnSF achieves speedups orders of magnitude over existing methods while maintaining high accuracy and robustness. We demonstrate the effectiveness of LD-EnSF on several challenging high-dimensional benchmarks with highly sparse (in both space and time) and noisy observations.

## 1 Introduction

Data assimilation (DA) (Sanz-Alonso et al., 2023) plays a central role in improving simulation accuracy for complex physical systems by integrating observational data into numerical models. It is extensively applied in real-world domains such as weather forecasting (Schneider et al., 2022), computational fluid dynamics (Carlberg et al., 2019), and sea ice modeling (Zuo et al., 2021), where systems exhibit intricate interactions and uncertainties. Classical Bayesian filtering methods, such as the Kalman Filter (Kalman, 1960), Ensemble Kalman Filter (EnKF) (Evensen, 1994) and particle filters (Künsch, 2013), are widely used because of their computational efficiency. Variants such as the Local Ensemble Transform Kalman Filter (LETKF) (Hunt et al., 2007) and adaptive sampling methods (Bishop et al., 2001) further extend the applicability of EnKF. However, standard EnKFs struggle to handle high-dimensional and nonlinear systems because of quadratic complexity in dimensionality and the linearized posterior assumption. Variational methods like 4D-Var (Rabier & Liu, 2003) offer improved accuracy but require complex optimization with repeated forward simulations, leading to high computational cost. Ensemble methods such as 4DEnVar (Desroziers et al., 2014) learn linear tangent models to accelerate the optimization process, but the approximation gap can be significant. Though data-driven methods such as Tensor-Var (Yang et al., 2025) aim to accelerate the 4DVar through kernelized representations, they still require solving complex optimization problems and differentiating through multi-step dynamics, making direct application to high-dimensional systems nontrivial.

To overcome these limitations, the Ensemble Score Filter (EnSF) (Bao et al., 2024b) has been developed for high-dimensional and nonlinear data assimilation, offering linear complexity and more accurate posterior approximation. Unlike the EnKF, EnSF encodes probability densities via score functions and generates samples by solving reverse-time stochastic differential equations (SDEs). However, it performs poorly under sparse observations, where the likelihood gradient is zero in the unobserved regions. Latent-EnSF (Si & Chen, 2025) addresses this issue by using Variational

---

*Corresponding author.

Autoencoders (VAEs) (Kingma & Welling, 2014) to project both states and observations into a shared latent space, where score-based filtering can be effectively applied. The latent representations enable more informative gradients, thus mitigating the limitations of sparsity. After assimilation, the system dynamics are evolved in the original full space using existing simulation methods. While this approach is flexible and model-agnostic, it remains computationally demanding because of the complexity of propagating full-space dynamics, limiting its applications to real-time data assimilation and resource-constrained settings.

Surrogate models, especially neural network-based approaches, offer a solution to this problem (Rudy et al., 2017; Champion et al., 2019; Lu et al., 2021; Li et al., 2020; Floryan & Graham, 2022; Vlachas et al., 2022; Bonev et al., 2023; Chen et al., 2024; Yue et al., 2025a;b; Qiu et al., 2025). These models aim to learn reduced representations of the system and directly evolve the dynamics in a low-dimensional latent space, thereby avoiding repeated calls to the original numerical simulator. Among these approaches, Latent Dynamics Networks (LDNets) (Regazzoni et al., 2024) outperform traditional surrogate models by achieving higher accuracy while using significantly fewer trainable parameters across several complex systems. It jointly learns a smooth latent representation and its temporal evolution without requiring a separate encoder. Crucially, LDNets evolve latent states conditioned on system parameters (e.g., initial conditions, system coefficients). This strong dependency means that inaccuracies in the parameters can lead to significant prediction errors. Yet it also provides an opportunity: by integrating data assimilation, these parameters can be estimated and corrected during inference, which makes them particularly suitable for data assimilation.

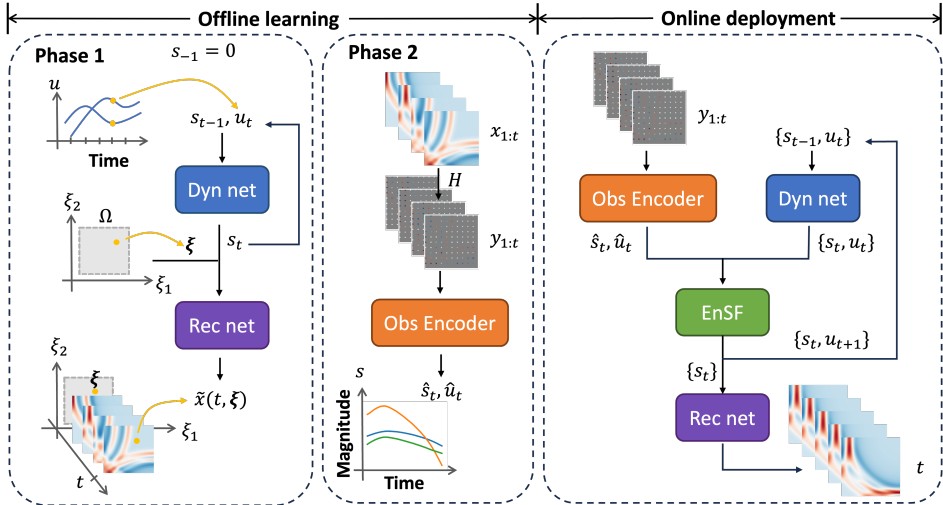

Figure 1: **The framework of the LD-EnSF method. Offline learning**: In phase 1, the LDNet is trained based on the dataset to capture the latent dynamics. In phase 2, an LSTM encoder is trained to encode the observation history $y_{1:t}$ matching the latent variable $s_t$ and parameter $u_t$ of the trained LDNet at time $t$. **Online deployment**: at each assimilation time step, the LD-EnSF assimilates an ensemble of prior latent pairs $\{s_t, u_t\}$ with LSTM encoded latent pairs $(\hat{s}_t, \hat{u}_t)$. The posterior latent states are then used to reconstruct the full states at arbitrary time and space points.

In this work, we accelerate the assimilation process by avoiding the costly forward propagation of full dynamics while addressing the limitations of EnSF regarding sparse observations. We introduce the **Latent Dynamics Ensemble Score Filter (LD-EnSF)**, see Fig. 1 for its general framework. Our approach leverages LDNets to model and preserve system dynamics within a very low-dimensional latent space where EnSF is applied. Furthermore, we incorporate a Long Short-Term Memory (LSTM) (Hochreiter, 1997) encoder for mapping observations to the latent space, enabling the efficient use of past observations, especially in scenarios with high observation sparsity. The code for data generation, model training, and inference can be found at `https://github.com/pengpeng-xiao/ld-ensf`. Our main contributions are summarized as follows:

- We propose LD-EnSF, an enhanced data assimilation model developed based on the Latent-EnSF for Bayesian filtering. We replace the disconnected VAE and forward propagation model with a more cohesive framework, using LDNets as an improved surrogate model

to perform the assimilation process in a low-dimensional latent space. This significantly reduces computational costs and enables real-time data assimilation with large ensembles.

- We advance LDNets with a novel initialization scheme, a two-stage training strategy, and improved network architectures, enabling high accuracy and low-dimensional representations of complex dynamics.

- We propose a new sparse observation encoder based on LSTM, designed to align both latent states and system parameters from LDNets. This encoder effectively captures historical and irregularly spaced observations, enabling accurate and robust joint data assimilation of states and parameters.

- We demonstrate the superior accuracy, efficiency, and robustness of LD-EnSF through high-dimensional data assimilation examples of increasing complexity, including Kolmogorov flow, tsunami modeling, and atmospheric modeling, with extreme spatial and temporal sparsity.

## 2 BACKGROUND

In this section, we introduce the basic concepts and problem setup for data assimilation in the context of filtering, and present the ensemble score filter (EnSF) and its latent space variant, Latent-EnSF.

### 2.1 PROBLEM SETUP

We denote $x_t \in \mathbb{R}^{d_x}$ as a $d_x$-dimensional state variable of a dynamical system at (discrete) time $t \in \mathbb{Z}^+$, with initial state $x_0$. Given the state $x_{t-1}$ at time $t-1$, with $t = 1, 2, \ldots$, the evolution of the state from $t-1$ to $t$ is modeled as

$$x_t = F(x_{t-1}, u_t), \tag{1}$$

where $F : \mathbb{R}^{d_x} \times \mathbb{R}^{d_u} \to \mathbb{R}^{d_x}$ is a non-linear forward map, and $u_t \in \mathbb{R}^{d_u}$ represents a $d_u$-dimensional uncertain parameter. By $y_t \in \mathbb{R}^{d_y}$ we denote $d_y$-dimensional noisy observation data, given as

$$y_t = H(x_t) + \gamma_t, \tag{2}$$

where $H : \mathbb{R}^{d_x} \to \mathbb{R}^{d_y}$ is the observation map, and $\gamma_t$ represents the observation noise.

Due to the model inadequacies and input parameter uncertainties, the model of the dynamical system in Eq. 1 may produce inaccurate predictions of the ground truth. The goal of data assimilation is to find the best estimate, denoted by $\hat{x}_t$, of the ground truth, given the observation data $y_{1:t} = (y_1, y_2, \cdots, y_t)$ up to time $t$. This requires us to compute the conditional probability density function of the state, denoted as $P(x_t|y_{1:t})$, which is often non-Gaussian. In the Bayesian filtering framework (Appendix A.1), data assimilation is formulated as a two-step process of prediction and update, where accurately approximating the prior and posterior densities remains a challenge. This also extends to inferring the uncertain input parameter $u_t$, which remains a challenging task under sparse or noisy observations.

### 2.2 ENSF AND LATENT-ENSF

Built on the recent advances in score-based sampling methods (Song et al., 2021), EnSF (Bao et al., 2024b) draws samples from the posterior $P(x_t|y_{1:t})$ using its score $\nabla_x \log P(x_t|y_{1:t})$ through Monte Carlo integration for the prior score in the prediction step, and a damped likelihood score in the update step; see details of the method and algorithm in Appendix A.2. EnSF utilizes the explicit likelihood function and diffusion process to generate samples without assuming system linearity, proving effective for nonlinear, high-dimensional systems like the Lorenz 96 model (Bao et al., 2024b) and quasi-geostrophic dynamics (Bao et al., 2024a). However, its performance is significantly hindered in sparse observation scenarios, where the score of the likelihood function vanishes in unobserved state components (Si & Chen, 2025).

To address the limitations of EnSF with sparse observations, Latent-EnSF (Si & Chen, 2025) performs data assimilation in a shared latent space, where both states $x_t$ and observations $y_t$ are encoded using a coupled variational autoencoder (VAE). The VAE, comprising a state encoder $\mathcal{E}_{\text{state}}$, an observation

encoder $\mathcal{E}_{\text{obs}}$, and a decoder $\mathcal{D}$, is trained to minimize a loss function balancing state-observation consistency, reconstruction errors, and latent distribution regularization. After training, Latent-EnSF applies EnSF in the latent space using encoded states $\mathcal{E}_{\text{state}}(x_t)$ and observations $\mathcal{E}_{\text{obs}}(y_t)$. Assimilated latent samples are decoded into the full state space and propagated via the dynamical model in Eq. 1. This method mitigates the vanishing score issue faced by EnSF for sparse observations and achieves high accuracy even under extreme sparsity, where the vanilla EnSF fails, such as using only $0.44\%$ of state components for a shallow water wave propagation problem.

While Latent-EnSF improves sampling efficiency over the vanilla EnSF, its forward evolution still requires numerical propagation of the full dynamical system, which is computationally prohibitive for large-scale and real-time applications. Moreover, Latent-EnSF's latent states exhibit oscillatory and non-smooth behavior, making it challenging to construct a stable dynamical model in the latent space approximating complex physical dynamics. These challenges motivate the method we develop in this work.

## 3 METHODOLOGY

In this section, we present LD-EnSF, a fast, robust, and accurate data assimilation method that avoids using full-model dynamics during assimilation. It accomplishes this by learning a latent representation of the system state via LDNets and effectively incorporating sparse observations using an LSTM encoder, which extends the VAE encoder in Latent-EnSF by accommodating not only historical observations but also irregular spatial sparsity.

### 3.1 LATENT DYNAMICS NETWORK

Latent Dynamics Network (LDNet) (Regazzoni et al., 2024) has been demonstrated as one of the most capable methods in capturing low-dimensional representations of many complex dynamics. The architecture consists of a dynamics network $\mathcal{F}_{\theta_1}$, which evolves the latent state $s_t$, and a reconstruction network $\mathcal{R}_{\theta_2}$, which maps latent states back to the full state space at any spatial point.

To extend and improve LDNet for learning more complex dynamics with varying initial conditions in the context of data assimilation, we propose three new variants: (1) shifting initial latent state, (2) two-stage training and fine-tuning, and (3) a new architecture of the reconstruction net.

The dynamics network $\mathcal{F}_{\theta_1}$, as shown in Fig. 1 offline learning phase 1, takes the latent state $s_{t-1} \in \mathbb{R}^{d_s}$ and parameter $u_t$ as input and outputs the time derivative of $s_{t-1}$:

$$\dot{s}_{t-1} = \mathcal{F}_{\theta_1}(s_{t-1}, u_t), \quad t = 0, 1, 2, \ldots . \tag{3}$$

The latent state $s_t$ is updated from the one-step forward Euler scheme as follows:

$$s_t = s_{t-1} + \Delta t \, \dot{s}_{t-1}, \quad t = 0, 1, 2, \ldots, \tag{4}$$

where $\Delta t$ is the time step, set as $\Delta t = T/n$ for $n$ steps. Unlike Regazzoni et al. (2024), we initialize the latent state as $s_{-1} = 0$ instead of $s_0 = 0$ to accommodate varying initial conditions.

The reconstruction network $\mathcal{R}_{\theta_2}$, also shown in Fig. 1, reconstructs from the latent state $s_t$ an approximate full state $\tilde{x}(t, \xi)$ at any spatial query point $\xi \in \Omega$ in domain $\Omega$ as:

$$\tilde{x}(t, \xi) = \mathcal{R}_{\theta_2}(s_t, \xi), \quad t = 0, 1, 2, \ldots . \tag{}$$

To train the LDNet, we propose a two-stage training strategy. First, we jointly train both the dynamics network and the reconstruction network by minimizing the loss:

$$\mathcal{L}(\theta_1, \theta_2) = \frac{1}{NMn} \sum_{j=1}^{N} \sum_{t} \sum_{\xi} \|\tilde{x}_j(t, \xi) - x_j(t, \xi)\|^2, \tag{5}$$

where $\{x_j\}_{j=1}^{N}$ are $N$ trajectories, and $M$ is the number of spatial points $\xi$ in each trajectory. In the second stage, we fine-tune the reconstruction network $\mathcal{R}_{\theta_2}$ with fixed latent representations from the dynamics network, effectively reducing reconstruction errors. This novel training strategy ensures accurate and efficient modeling of both the latent dynamics and the full state reconstruction.

Moreover, to enhance the reconstruction power for more complex dynamics, we propose to employ a ResNet-based architecture (He et al., 2016) and integrate Fourier encoding (Tancik et al., 2020; Qiu et al., 2024; Salvador & Marsden, 2024) to better capture high-frequency spatial components, defined as $\xi \mapsto [\cos B\xi, \sin B\xi]$, where $B \in \mathbb{R}^{m \times d_\xi}$ is a trainable parameter matrix, and $m$ is the hyperparameter controlling the dimensionality of the encoding.

## 3.2 Observation encoding

To map the observations into the learned latent space, we propose training a separate observation encoder employing an LSTM (Hochreiter, 1997) to address the challenge of encoding sparse and noisy observations. In contrast to the VAE encoder used in Latent-EnSF, which constructs latent representations of the sparse observations at step $t$, the LSTM encoder leverages temporal correlations in sequential observations $y_{1:t}$, effectively learning a nonlinear time-delay embedding (Noakes, 1991; Takens, 2006). In addition, while the VAE in Latent-EnSF can only handle observations on a regular grid, the LSTM encoder is capable of effectively handling observations at random/irregular locations.

As shown in Fig. 1, the LSTM encoder, denoted as $\mathcal{E}_{\theta_3} : \mathbb{R}^{d_y \times t} \to \mathbb{R}^{d_u + d_s}$, is parameterized by trainable parameters $\theta_3$, with historical observations $y_{1:t}$ up to time $t$ as input. The output of the LSTM network is a pair consisting of the approximate latent state $\hat{s}_t \in \mathbb{R}^{d_s}$ and parameter $\hat{u}_t \in \mathbb{R}^{d_u}$:

$$(\hat{s}_t, \hat{u}_t) = \mathcal{E}_{\theta_3}(y_{1:t}), \tag{6}$$

where $y_t = H(x_t)$ is the noiseless sparse observation at time $t$. This encoder facilitates the assimilation of not only the state as in Latent-EnSF but also the parameter $u_t$.

To train the LSTM encoder, we minimize the following loss:

$$\mathcal{L}(\theta_3) = \frac{1}{Nn} \sum_{j=1}^{N} \sum_{t} \left( \left\| \hat{s}_t^{(j)} - s_t^{(j)} \right\|^2 + \left\| \hat{u}_t^{(j)} - u_t^{(j)} \right\|^2 \right), \tag{7}$$

where the $N$ trajectories are sampled from the LDNet training data, with $s_t$ provided by the trained LDNet. This loss enforces the alignment between the encoded observations and the corresponding latent states and parameters, enabling additional parameter estimation during data assimilation. While proper weighting between these two terms can be explored, we found that common normalization or standardization of the states and parameters to similar ranges works well without weighting.

## 3.3 LD-EnSF: Latent dynamics ensemble score filter

After training the LDNet and LSTM offline, we integrate EnSF to perform data assimilation in the latent space, as illustrated in the online deployment phase in Fig. 1. Let $\kappa_t = (s_t, u_t)$ denote the augmented latent state. The corresponding latent observation encoded by the LSTM network in Eq. 6 is given by $\phi_t = (\hat{s}_t, \hat{u}_t)$. The latent observation data can be approximately modeled as

$$\phi_t = H_{\text{latent}}(\kappa_t) + \hat{\gamma}_t, \tag{8}$$

where we take the identity map $H_{\text{latent}}(\kappa_t) = \kappa_t$ and estimate latent observation noise $\hat{\gamma}_t$ via LSTM encoding of the true observation noise $\gamma_t$ from Eq. 2, as detailed in Appendix E.1. To this end, the Bayesian filtering problem in latent space is formulated in two steps. In the prediction step, we have

$$P(\kappa_t | \phi_{1:t-1}) = \int P(\kappa_t | \kappa_{t-1}) P(\kappa_{t-1} | \phi_{1:t-1}) d\kappa_{t-1}, \tag{9}$$

with the transition probability $P(\kappa_t | \kappa_{t-1})$ derived from the latent dynamics in Eqs. 3 and 4, where we draw samples of $u_t$ from the empirical posterior of $u_{t-1}$. In the update step, we have

$$P(\kappa_t | \phi_{1:t}) \propto P(\phi_t | \kappa_t) P(\kappa_t | \phi_{1:t-1}), \tag{10}$$

with the likelihood function $P(\phi_t | \kappa_t)$ defined through the latent observation model (Eq. 8). We can then employ EnSF to solve this Bayesian filtering problem in the latent space, with the latent dynamics (LD) evolved as a surrogate of the full dynamics. We present one step of the LD-EnSF in Algorithm 1.

As shown in Fig. 1, the assimilation process runs entirely in the latent space, removing the need to map back to the full state space during assimilation. Once latent states are obtained, the reconstruction network can be used to recover the full system state at any desired spatial location. Moreover, the smoothness of the latent trajectories, as we will show in our experiments in Section 4.1, enables accurate interpolation in time, allowing the full state to be evaluated at any continuous time point.

---

**Algorithm 1** One step of LD-EnSF

---

**Input:** Ensemble of the latent states and parameters $\{\kappa_{t-1}\}$ from distribution $P(\kappa_{t-1}|\phi_{1:t-1})$ and the observations $y_{1:t}$. Dynamics network $\mathcal{F}_{\theta_1}$, observation encoder $\mathcal{E}_{\theta_3}$.
**Output:** Ensemble of the latent states and parameters $\{\kappa_t\}$ from posterior distribution $P(\kappa_t|\phi_{1:t})$.

1: Simulate the latent dynamics in Equations 3 and 4 from the ensemble of samples $\{\kappa_{t-1}\}$ to obtain samples $\{\kappa_t\}$ following $P(\kappa_t|\phi_{1:t-1})$, where we draw samples of $\{u_t\}$ based on $\{u_{t-1}\}$.
2: Encode the historical observations by the LSTM network as $\phi_t = \mathcal{E}_{\theta_3}(y_{1:t})$.
3: Draw an ensemble of random samples $\{\kappa_{t,1}\}$ from the standard normal distribution $N(0, I)$.
4: Run the reverse-time SDE of EnSF from each $\kappa_{t,1}$ to obtain the posterior sample $\kappa_t = \kappa_{t,0}$.

---

## 4 EXPERIMENTS

We consider three experimental examples: (1) a chaotic system modeled by Kolmogorov flow with uncertain viscosity, (2) shallow water wave propagation in tsunami modeling with uncertain initial conditions (for earthquake locations), and (3) forced hyperviscous rotation on a sphere with earth-like topography in atmospheric modeling with an uncertain forcing term. Below, we briefly present these problems, with detailed equations and the setup for data generation provided in Appendix B.

**Kolmogorov flow.** Kolmogorov flow is a canonical turbulent system driven by a sinusoidal body force, parameterized by an uncertain Reynolds number $Re$ (Kochkov et al., 2021). The spatial resolution is set as $150 \times 150$. We burn in for 100 time steps, then simulate 200 time steps with a step size of $\delta t = 0.04$. Observations are provided every 5 steps on a regular $10 \times 10$ grid (0.44% of the domain) and on 100 randomly placed points for evaluating LD-EnSF, resulting in 40 assimilation steps.

**Tsunami modeling.** We use simplified shallow water equations for tsunami modeling, where the initial condition is specified as a Gaussian bump randomly placed in the spatial domain, following the setup in Si & Chen (2025). For the simulation, we discretize the spatial domain into a uniform grid of $150 \times 150$. The simulation is carried out over $2{,}000$ time steps with a time step size of $\delta t \approx 21$ seconds. Observation data are provided on a regular $10 \times 10$ grid (and 100 random/irregular points to test LD-EnSF) for every 40 simulation steps, leading to 50 assimilation steps.

**Atmospheric modeling.** This setup is adapted from the planetswe example in the Well (Ohana et al., 2024). The spatial resolution is set as $512 \times 256$. We simulate for half a year (21 days as we set 42 days for a year), totaling 504 hours for approximately 30,240 time steps with an adaptive time step size of about $\delta t \approx 60$ seconds, where the initial condition is derived from the 500 hPa pressure level of the reanalysis dataset ERA5 (Hersbach et al., 2020). Observations are provided on a regular $16 \times 8$ grid for every 8 hours of simulation, leading to 63 assimilation steps, featuring extreme spatial ($\sim$0.1%) and temporal ($\sim$0.2%) sparsity.

### 4.1 OFFLINE LEARNING OF LDNETS

LDNets are trained via hyperparameter search (Biewald, 2020) with latent states discretized at observation times. The hyperparameter selection and normalization method are detailed in Appendix C.1. In our current experimental settings, we use a static $u(t)$. LDNet can also accommodate slowly varying $u(t)$ without modification. If $u_t$ changes more drastically, an explicit dynamical model $u_{t+1} = F_u(u_t)$ would need to be learned in addition to the latent state dynamics. The test error of LDNet as a surrogate model is reported in Table 1, compared with the VAE used in Latent-EnSF (Si & Chen, 2025) with a Latent Diffusion Model (Rombach et al., 2022) architecture. For a fair comparison, we also construct a latent dynamical model by integrating the VAE with an LSTM (referred to as VAE-dyn) jointly trained to predict the latent state as $\mathcal{E}_{\text{state}}(x_{t+1}) = \text{LSTM}(\mathcal{E}_{\text{state}}(x_0), \ldots, \mathcal{E}_{\text{state}}(x_t))$. This setup allows the LSTM to propagate the initial latent state. As shown in Fig. 3, while the VAE effectively learns latent representations of the full dynamics and achieves low reconstruction error (with VAE-dyn exhibiting similar state reconstruction error), the LSTM in VAE-dyn fails to maintain stable long-term latent predictions, leading to rapid accumulation of reconstruction errors over time. In contrast, our LDNet achieves the lowest approximation errors in all three examples. Note that the original LDNet fails to capture the varying initial conditions in the tsunami example (resulting

in large errors), while our improved LDNet accommodates this. Furthermore, the original LDNet exhibits much larger errors than ours in capturing high-frequency dynamics for the Kolmogorov flow.

Table 1: Relative RMSE (averaged in time) for the approximation of full dynamics by LDNet (original vs ours), VAE reconstruction, and VAE-dyn (with LSTM for latent dynamics).

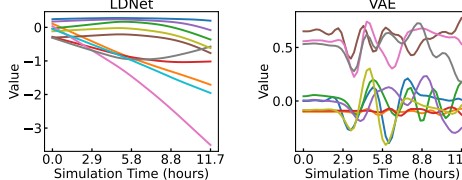

| Example | VAE | VAE-dyn | LDNet (original) | LDNet (ours) |
|---|---|---|---|---|
| Kolmogorov | 0.0131 | 0.964 | 0.0349 | **0.0123** |
| Tsunami | 0.0309 | 1.33 | 0.1837 | **0.0168** |
| Atmospheric | 0.0856 | 0.483 | 0.1042 | **0.0656** |

Figure 2: Smoother latent states of LDNet (left) than those of VAE (right) for tsunami modeling.

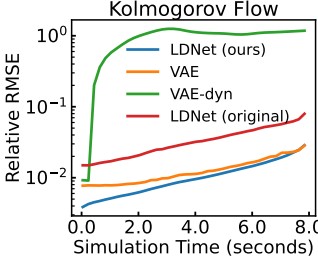 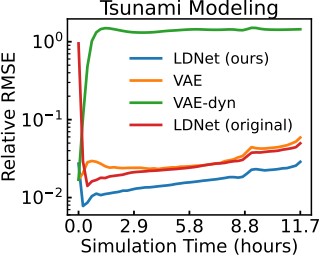 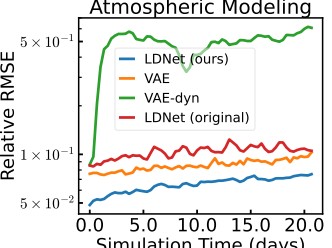

Figure 3: Relative RMSE for the full-state approximation by LDNet (original and ours), VAE reconstruction, and VAE-dyn (with LSTM trained to predict the VAE latent dynamics).

Compared to the latent states of VAE, the latent states of LDNet are noticeably smoother, as shown in Fig. 2 for the tsunami example, and in Fig. 6 in Appendix D for other examples. The smoother latent states make it much easier for the latent observations (mapped from the sparse observations by an observation encoder) to match the predicted latent states. This can help to enhance the data assimilation accuracy of the LDNet-based LD-EnSF compared to the VAE-based Latent-EnSF, supported by results in Section 4.3. Moreover, the smooth latent states of LDNet facilitate accurate temporal interpolation, allowing for the reconstruction of full states at arbitrary times besides those at observation times (Fig. 7 in Appendix D).

## 4.2 OFFLINE LEARNING OF OBSERVATION ENCODER

After training LDNets, we generate a sequence of the latent states $s_t$ in Eq. 4 for each input parameter sequence $u_t$ by running the latent dynamical model (Eq. 3). The observation operator $H$ is set as a sparse sub-sampling matrix that selects the state values from grid points. The observational training data at this stage do not include observation noise. Using single-layer LSTMs, we are able to achieve an average test error of $0.07\%$ for Kolmogorov flow, $0.5\%$ for tsunami modeling, and $2.53\%$ for atmospheric modeling. Detailed training configurations are provided in Appendix C.1.

**Irregular observations.** For a fair comparison with the VAE encoder in Latent-EnSF, we use sparse observations on equidistant grid points in the following experiments. However, our LSTM encoder can also effectively handle arbitrarily located observations, which is crucial in practical applications. In Appendix E.4, we provide robust and accurate assimilation results using the LSTM encoder with 100 irregularly sampled points for Kolmogorov and tsunami, and 128 for the atmospheric case.

## 4.3 ONLINE DEPLOYMENT OF LD-ENSF

To deploy the trained LDNets and LSTM encoder for online data assimilation, we initialize an ensemble of 20 samples $\{s_{-1}, u_0\}$, each comprising the latent state $s_{-1} = 0$ and parameter $u_0$ for the assimilation. For the Kolmogorov flow, $u_0$ represents the Reynolds number $Re$ and is randomly drawn from a uniform distribution $\mathcal{U}([500, 1500])$. For the tsunami modeling, $u_0 \in \mathcal{U}([0, 0.5], [0, 0.5])$ represents the coordinates of the local Gaussian bump for initial surface elevation, which is randomly sampled in the upper-left quarter of the domain. In the atmospheric modeling experiment, we define the time-dependent forcing using $u_0 \in \mathcal{U}([0.1, 30], [1, 4])$, where the two components represent

the amplitude $h_f^0$ and latitudinal spread $\sigma$ of a seasonally and diurnally shifting hotspot on the geopotential field. The assimilation process for the complex variation of the dynamics is visualized in Appendix H.

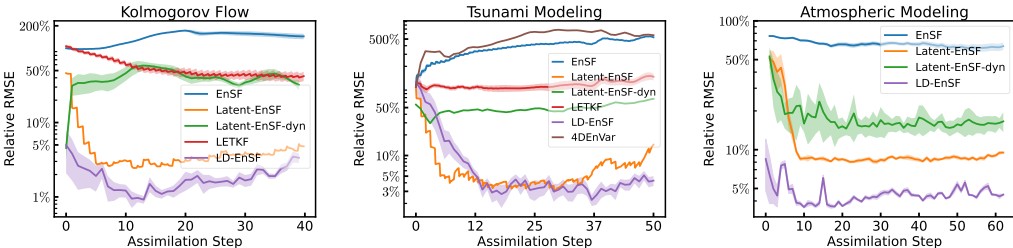

Figure 4: Relative RMSE with uncertainty estimate of the assimilation results for the three examples. LETKF is not shown in the atmospheric modeling case because of divergence.

**Comparison of assimilation accuracy.** We compare the assimilation accuracy of EnSF, Latent-EnSF (by VAE), Latent-EnSF-dyn (by VAE-dyn), LETKF (a state-of-the-art variant of the EnKF (Hunt et al., 2007)), 4DEnVar (for tsunami modeling), and LD-EnSF with $10\%$ observation noise for all examples, as shown in Fig. 4. We can observe that the full-space methods LETKF and especially EnSF fail to assimilate highly sparse and noisy observations in the high-dimensional setting. In particular, the EnSF fails because of noninformative gradients, which is resolved by the modifications made in the latent space methods. Latent-EnSF-dyn, which uses VAE-dyn as a surrogate forward model, has decreased performance compared with Latent-EnSF because of limited forecast accuracy. LD-EnSF achieves the smallest assimilation errors, as also visualized in Fig. 5 for the atmospheric example at the final assimilation step. For the extremely sparse observational data in this example, 0.1% data in space and $0.2\%$ in time, our LD-EnSF still preserves high assimilation accuracy with around 5% relative RMSE for 10% observation noise, while LETKF gives rise to numerical instability issues (to satisfy the CFL condition) that stopped the assimilation early (not reported in the figure).

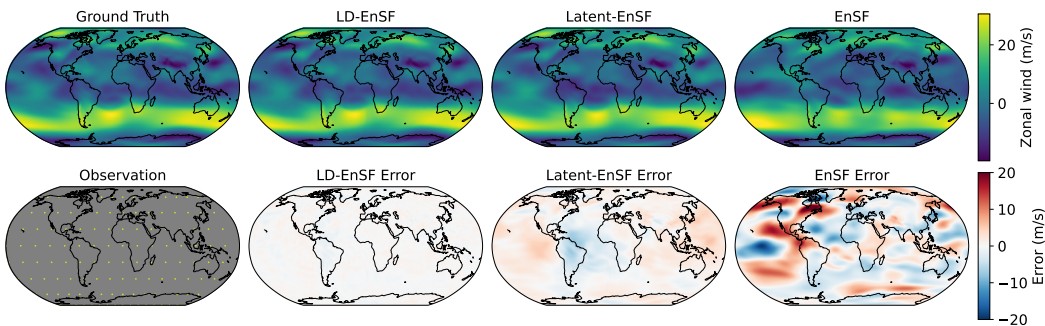

Figure 5: Zonal wind velocity field at the 500 hPa pressure level from an atmospheric simulation with a given forcing field. Top-left: ground truth; bottom-left: sparse observations. Remaining columns: assimilation results (top) and corresponding errors (bottom) at the final time step.

**Comparison of computational cost.** We evaluate the computational efficiency of LD-EnSF compared to other methods. EnSF, Latent-EnSF, and LETKF all require simulating the full dynamics during data assimilation, whereas LD-EnSF evolves only surrogate latent dynamics. As shown in Table 2, compared to the full dynamics used by other methods, evolving latent dynamics by LD-EnSF achieves speedups of $2 \times 10^5$, $4 \times 10^3$, and $5 \times 10^5$ times ($T_d$) for the three examples. As LD-EnSF performs all data assimilation steps in the latent space, it does not require transforming assimilated latent states back to full states at each assimilation step. In contrast, Latent-EnSF reconstructs or decodes the latent states to full states at every step, incurring additional reconstruction time ($T_r$). Additionally, LDNet learns a much lower-dimensional latent representation (10, 12, and 52 dimensions compared to 400, 400, and 512 dimensions in Latent-EnSF), which further reduces the assimilation time ($T_f$). Meanwhile, LETKF incurs significantly higher computational costs because of its assimilation process. The high efficiency of LD-EnSF not only enables real-time data assimilation and the use of larger ensembles to capture extreme events, but also provides increasing advantages in online cost as

the number of assimilation cycles grows. A detailed comparison of online gains versus offline cost is given in Appendix C.2.

Table 2: Comparison in wall-clock runtime in seconds. We denote the time for evolution of the dynamics as $T_d$, the time for data assimilation as $T_f$, and the time for reconstructing the full state from the latent state as $T_r$ (only needed at the last time step for LD-EnSF). The assimilation dimension is denoted as $D_s$. All results are for the full trajectories of an ensemble of 100 samples. The device is AMD 7543 CPU by default (64 processors for parallel simulation in atmospheric modeling), unless GPU (a single NVIDIA RTX A6000 GPU) is specified.

| Example | Kolmogorov Flow | | | | Tsunami Modeling | | | | Atmospheric Modeling | | | |
|---|---|---|---|---|---|---|---|---|---|---|---|---|
| **Metric** | LETKF | EnSF | Latent-EnSF | LD-EnSF | LETKF | EnSF | Latent-EnSF | LD-EnSF | LETKF | EnSF | Latent-EnSF | LD-EnSF |
| $T_d$ (s) | 10,829 | 10,829 | 10,829 | **0.049** | 211.30 | 211.30 | 211.30 | **0.050** | 35,603 | 35,603 | 35,603 | **0.070** |
| $T_f$ (s) | 12,729 | 40.13 | 0.71 | **0.35** | 10,440 | 83.86 | 0.66 | **0.37** | 69,820 | 27.36 | 2.42 | **1.42** |
| $T_r$ (GPU) (s) | – | – | 6.86 | **0.0018** | – | – | 5.11 | **0.0014** | – | – | 35.08 | **0.017** |
| dimension $D_s$ | 45,000 | 45,000 | 400 | **10** | 67,500 | 67,500 | 400 | **12** | 393,216 | 393,216 | 512 | **52** |

**Robustness and sensitivity of LD-EnSF.** Note that we construct the LSTM encoder without observation noise. To assess the robustness of LD-EnSF to observation noise, we perform data assimilation with varying noise levels (no noise, 5%, 10%, and 20%) for all examples. The relative RMSE of assimilated full states, parameters, and latent states is shown in Appendix E.5. While the assimilation errors increase with noise, the increase remains modest, and the errors drop significantly in the initial phase. This demonstrates the robustness of LD-EnSF, achieving high accuracy for both the states and the parameters (averaged over 20 experiments) despite highly sparse and noisy observations. In addition, we examine the effect of non-Gaussian and heteroskedastic noise in Appendix E.2, and evaluate out-of-distribution generalization in Appendix E.3. Sensitivity and stability analyses with respect to latent dimension, ensemble size, and observation sparsity are presented in Appendix F.

**Additional ablation studies**. We present additional ablations in Appendix G, systematically evaluating the architectural choices of the LDNet surrogate and the observation encoder, their training strategies, and the impact of their integrated latent-space assimilation. These results isolate each component's contribution and provide concrete support for the design choices in LD-EnSF.

## 5 RELATED WORK

**Score-based data assimilation:** Score-based methods have emerged as promising tools for nonlinear data assimilation. Bao et al. (2024b) introduced a filter that integrates diffusion models into Bayesian filtering for state estimation, while Bao et al. (2024a) proposed a training-free ensemble score estimation method that has been successfully applied to geophysical systems. Rozet & Louppe (2023) used conditional score-based generative models to reconstruct entire trajectories; however, these smoothing approaches assume access to future observations and thus differ from the real-time filtering required in practical scenarios, where only past and present data are available. Chen et al. (2025) extended Rozet & Louppe (2023) by incorporating stochastic interpolants and adapting the framework for filtering. However, their surrogate model relies solely on score-based predictions, and their dynamics are propagated in the full-state space, resulting in significantly higher computational costs compared to latent-space approaches. Huang et al. (2024) proposed a conditional generation method, but its performance degrades after several assimilation steps, with a simple interpolation baseline outperforming it, indicating limitations in handling states that deviate substantially from the training distribution. They also rely on expensive score-function training and full-state generative modeling, contrasting with the training-free EnSF that we utilize. More recent work includes the state-observation augmented diffusion (SOAD) model (Li et al., 2024c) and sequential Langevin sampling (Ding et al., 2024), both of which require expensive training of the score function and differ from the training-free EnSF that we build upon in this work.

**Latent space assimilation:** Latent-space methods have been shown to improve both the accuracy and efficiency of data assimilation (Peyron et al., 2021; Bachlechner et al., 2021; Penny et al., 2022; Pawar & San, 2022; Cheng et al., 2023; Mücke et al., 2024). For instance, Chen et al. (2023) integrated Feedforward Neural Networks (FNN) with the EnKF to model latent dynamics, while Li et al. (2024b) proposed using Spherical Implicit Neural Representations (SINR) and Neural ODEs (Chen et al., 2018) with the EnKF. These approaches mainly focus on Kalman-based filtering, while our work takes a different direction. We extend score-based EnSF to handle sparse observations by introducing

a new LSTM observation encoder, while maintaining high computational efficiency through latent-space assimilation with LDNets. Deep Bayesian filtering (DBF) Tarumi et al. (2024) also performs filtering in a learned latent space, but unlike LD-EnSF it trains an end-to-end generative model with linear latent transitions and a Gaussian inverse observation operator, rather than pairing a learned surrogate with a training-free EnSF update. Meanwhile, Frion et al. (2024) performs DA with a learned Koopman prior, and Singh et al. (2024b) links a Koopman-style latent evolution with an online DA loop. Similar to LD-EnSF, these method also operate in a compressed latent space.

**Deep learning and data assimilation:** A growing line of work integrates deep learning with classical variational and ensemble-based data assimilation. Examples include VAE-enhanced variational methods (Xiao et al., 2025b;c) and parameter-efficient hybrid EnVar systems (Xiao et al., 2025a). Gottwald & Reich (2021) couple delay-coordinate surrogate modeling with the EnKF. Singh et al. (2024a) introduce a recursive operator framework for semilinear PDEs that supports both forecasting and DA. In parallel, several learned 4D-Var systems extend classical variational DA. Examples include LEVDA (Si & Chen, 2026), Tensor-Var (Yang et al., 2025), 4DVarNets (Fablet et al., 2023; Beauchamp et al., 2023b), their ensemble and uncertainty-aware variants (Beauchamp et al., 2023a), fast attention-based DA surrogates (Wang et al., 2024), and large-scale hybrids (Li et al., 2024a).

## 6 CONCLUSION AND FUTURE WORK

In this work, we developed LD-EnSF, a robust, efficient, and accurate method for high-dimensional Bayesian data assimilation in large-scale dynamical systems with sparse and noisy observations. By integrating LDNets with improved initialization, training, and architectures, alongside LSTM-based historical observation encoding, LD-EnSF achieves a smooth, low-dimensional latent representation while enabling fast latent-space dynamics evolution and joint assimilation of state variables and system parameters. Numerical experiments on three challenging examples demonstrated its superior accuracy and efficiency compared to LETKF, EnSF, and Latent-EnSF.

For more complex uncertain system parameters, such as high-dimensional stochastic processes and spatially varying random fields, future work should explore effective strategies for encoding these into low-dimensional latent representations. For more complex dynamical systems that are difficult to capture with latent dynamics over long time horizons, our framework could be extended by iteratively applying LD-EnSF over shorter intervals, with the latent model adaptively retrained at each stage.

## ACKNOWLEDGMENTS

This work was supported in part by the National Science Foundation under Grant No. 2325631, 2245111, as well as 2025 IDEaS + Cloud Hub at Georgia Tech, with support from Microsoft. We acknowledge helpful discussions with Hanqing Wang.

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

## A   ADDITIONAL BACKGROUND

### A.1   BAYESIAN FILTER FRAMEWORK

In the Bayesian filter framework (Dore et al., 2009), the data assimilation problem becomes evolving $P(x_{t-1}|y_{1:t-1})$ to $P(x_t|y_{1:t})$ from time $t-1$ to $t$. This includes two steps: a prediction step followed by an update step. In the prediction step, we predict the density of $x_t$, denoted as $P(x_t|y_{1:t-1})$, from $P(x_{t-1}|y_{1:t-1})$ and the forward evolution of the dynamical model in Eq. 1 as

$$P(x_t|y_{1:t-1}) = \int P(x_t|x_{t-1})P(x_{t-1}|y_{1:t-1})dx_{t-1}, \tag{11}$$

where $P(x_t|x_{t-1})$ represents a transition probability. In the update step, given the new observation data $y_t$, the prior density $P(x_t|y_{1:t-1})$ from the prediction is updated to the posterior density $P(x_t|y_{1:t})$ by Bayes' rule as

$$P(x_t|y_{1:t}) = \frac{1}{Z}P(y_t|x_t)P(x_t|y_{1:t-1}). \tag{12}$$

Here, $P(y_t|x_t)$ is the likelihood function of the observation data $y_t$ determined by the observation model in Eq. 2, and $Z$ is the model evidence or the normalization constant given as $Z = \int P(y_t|x_t)P(x_t|y_{1:t-1})dx_t$, which is often intractable to compute.

### A.2   ENSEMBLE SCORE FILTER

A comparison between previous score-based assimilation methods and our LD-EnSF is shown in Table 3.

Table 3: Comparison of EnSF, Latent-EnSF, and LD-EnSF.

| Methods | Sparse Observations | Dynamics |
|---|---|---|
| EnSF | ✗ | ✗ |
| Latent-EnSF | ✓ | ✗ |
| LD-EnSF | ✓ | ✓ |

In EnSF, at a given physical time $t$, we define a pseudo-diffusion time $\tau \in \mathcal{T} = [0, 1]$, at which we progress

$$dx_{t,\tau} = f(x_{t,\tau}, \tau)d\tau + g(\tau)dW, \tag{13}$$

driven by a $d_x$-dimensional Wiener process $W$. Here, we use $x_{t,\tau}$ to indicate the state at physical time $t$ and diffusion time $\tau$. The drift term $f(x_{t,\tau}, \tau)$ and the diffusion term $g(\tau)$ are chosen as

$$f(x_{t,\tau}, \tau) = \frac{d\log\alpha_\tau}{d\tau}x_{t,\tau}, \ g^2(\tau) = \frac{d\beta_\tau^2}{d\tau} - 2\frac{d\log\alpha_\tau}{d\tau}\beta_\tau^2, \tag{14}$$

with $\alpha_\tau = 1 - \tau(1 - \epsilon_\alpha)$ and $\beta_\tau^2 = \tau$, where $\epsilon_\alpha$ is a small positive parameter to avoid $d\log\alpha_\tau/d\tau$ being undefined at $\tau = 1$ (e.g., $\epsilon_\alpha = 0.01$ as in our experiments). This choice leads to the conditional Gaussian distribution

$$x_{t,\tau}|x_{t,0} \sim \mathcal{N}(\alpha_\tau x_{t,0}, \beta_\tau^2 I), \tag{15}$$

which gradually transforms the data distribution taken as $x_{t,0} = x_t \sim P(x_t|y_{1:t})$ at $\tau = 0$ close to a standard normal distribution at $\tau = 1$. This transformation process can be reversed by integrating an SDE from $\tau = 1$ to $\tau = 0$ as

$$dx_{t,\tau} = [f(x_{t,\tau}, \tau) - g^2(\tau)\nabla_x \log P(x_{t,\tau}|y_{1:t})]d\tau + g(\tau)d\bar{W}, \tag{16}$$

where $\bar{W}$ is another Wiener process independent of $W$, and $\nabla_x \log P(x_{t,\tau}|y_{1:t})$ is the score of the density $P(x_{t,\tau}|y_{1:t})$ with the gradient $\nabla_x$ taken with respect to $x_{t,\tau}$. By this formulation, $x_{t,\tau}$ follows the same distribution with density $P(x_{t,\tau}|y_{1:t})$ in the forward and reverse-time SDEs.

To compute the score $\nabla_x \log P(x_{t,\tau}|y_{1:t})$ in Eq. 16, the EnSF (Bao et al., 2024b) uses

$$\begin{aligned}\nabla_x \log P(x_{t,\tau}|y_{1:t}) = &\nabla_x \log P(x_{t,\tau}|y_{1:t-1})\\ &+ h(\tau)\nabla_x \log P(y_t|x_{t,\tau}),\end{aligned} \tag{17}$$

where the damping function $h(\tau) = 1 - \tau$ is chosen to monotonically decrease in $[0, 1]$, with $h(1) = 0$ and $h(0) = 1$. The likelihood function $P(y_t|x_{t,\tau})$ in the second term can be explicitly derived from the observation map in Eq. 2. The first term can be approximated using a Monte Carlo approximation, with samples drawn from the distribution $P(x_{t,0}|y_{1:t-1})$.

With the score function $\nabla_x \log P(x_{t,\tau}|y_{1:t})$ evaluated as in Eq. 17, the samples from the target distribution $P(x_t|y_{1:t})$ can be generated by first drawing samples from $\mathcal{N}(0, I)$ and then solving the reverse-time SDE in Eq. 16 using, e.g., the Euler-Maruyama scheme. The workflow for a single-step data assimilation using the EnSF is summarized in Algorithm 2 in Appendix A.2.

The algorithm of EnSF is presented in Algorithm 2, where state samples are evolved forward according to the physical model and then refined via score-based sampling in the reverse-time process.

---

**Algorithm 2** One step of EnSF

---

**Input:** Ensemble of the states $\{x_{t-1}\}$ from distribution $P(x_{t-1}|y_{1:t-1})$ and new observation $y_t$.
**Output:** Ensemble of the states $\{x_t\}$ from distribution $P(x_t|y_{1:t})$.

1: Simulate the forward model in Equation 1 from $\{x_{t-1}\}$ to obtain samples $\{x_{t,0}\}$ following $P(x_t|y_{1:t-1})$.
2: Generate random samples $\{x_{t,1}\}$ from standard normal distribution $N(0, I)$.
3: Solve the reverse-time SDE in Equation 16 starting from samples $\{x_{t,1}\}$ using the score in Equation 17 to obtain $\{x_t\}$.

---

# B    DETAILS OF EXPERIMENTS SETTINGS

## B.1    TSUNAMI MODELING

We consider the shallow water equations for tsunami modeling, which are widely used to model the propagation of shallow water where the vertical depth of the water is much smaller than the horizontal scale. These equations are frequently applied in oceanographic and atmospheric fluid dynamics. In this study, we adopt a simplified form of tsunami modeling:

$$\frac{d\mathbf{v}}{dt} = -f\mathbf{k} \times \mathbf{v} - g\nabla\eta,$$
$$\frac{d\eta}{dt} = -\nabla \cdot ((\eta + H)\mathbf{v}), \tag{18}$$

where $\mathbf{v}$ is the two-dimensional velocity field, and $\eta$ represents the surface elevation. Both $\mathbf{v}$ and $\eta$ constitute the system states to be assimilated. Here, $H = 100$m denotes the mean depth of the fluid, $f$ is the full latitude varying Coriolis parameter, and $g$ is the constant representing gravitational acceleration. $\mathbf{k}$ refers to the unit vector in the vertical direction. We define a two-dimensional domain of size $L \times L$, where $L = 10^6$m in each direction. The initial condition is a displacement of the surface elevation modeled by a Gaussian bump with its center randomly and uniformly distributed in the lower-left quarter of the domain. The boundary conditions are set such that $\mathbf{v} = 0$. Over time, the wave dynamics become increasingly complex due to reflections at the boundaries. The spatial domain is discretized into a uniform grid of $150 \times 150$, following the setup of (Si & Chen, 2025). The simulation is carried out over 2000 time steps using an upwind scheme with a time step size of $\delta t \approx 21$ seconds. Including the initial condition, the dataset comprises 2001 time steps. We generate 200 trajectories, dividing them into training (60%), validation (20%), and evaluation (20%) sets.

## B.2    KOLMOGOROV FLOW

In the second example, we consider the Kolmogorov flow with an uncertain Reynolds number, a parametric family of statistically stationary turbulent flows driven by a body force. This incompressible fluid is governed by the Navier-Stokes equations (Kochkov et al., 2021):

$$\frac{d\mathbf{v}}{dt} = -\mathbf{v} \cdot \nabla\mathbf{v} + \frac{1}{Re}\nabla^2\mathbf{v} - \frac{1}{\rho}\nabla p + \mathbf{f},$$
$$\nabla \cdot \mathbf{v} = 0. \tag{19}$$

The external forcing term $\mathbf{f}$ is defined as $\mathbf{f} = \sin(4\xi_2)\hat{\boldsymbol{\xi}}_1 - 0.1\mathbf{v}$, where $\boldsymbol{\xi} = (\xi_1, \xi_2)$ is the spatial coordinate, $\hat{\boldsymbol{\xi}}_1 = (1,0)$, $\mathbf{v}$ is the velocity field, $p$ is the pressure field, and $\rho = 1$ denotes the fluid density. The fluid velocity $\mathbf{v}$ is the state variable to be assimilated. The spatial domain is defined as $[0, 2\pi]^2$ with periodic boundary conditions and a fixed initial condition. The flow complexity is controlled by the Reynolds number $Re$. The spatial resolution is set to $150 \times 150$. We simulate the flow over 300 time steps with a step size of $\delta t = 0.04$ and take the data from time steps 100 to 300. A total of 200 trajectories are generated, with the Reynolds number $Re$ randomly sampled from the range $[500, 1500]$. These trajectories are divided into training (60%), validation (20%), and evaluation (20%) sets.

### B.3 ATMOSPHERIC MODELING

In the third example, we model the atmosphere by adopting the PlanetSWE formulation from Ohana et al. (2024). This system serves as a simplified approximation of the primitive equations used in atmospheric modeling at a single pressure level.

$$
\begin{aligned}
\frac{\partial \mathbf{v}}{\partial t} &= -\mathbf{v} \cdot \nabla \mathbf{v} - g\nabla h - \nu\nabla^4 \mathbf{v} - 2\Omega \times \mathbf{v}, \\
\frac{\partial h}{\partial t} &= -H\nabla \cdot \mathbf{v} - \nabla \cdot (h\mathbf{v}) - \nu\nabla^4 h + F,
\end{aligned}
$$

where $\mathbf{v}$ is the two-dimensional velocity field, $h$ denotes the deviation of the pressure surface height from the mean $H$, and $\Omega$ represents the Coriolis parameter. The term $\nabla^4$ denotes hyperviscosity. The initial conditions are derived from the 500 hPa pressure level in the ERA5 reanalysis dataset. $F$ is an external forcing term that introduces both daily and annual seasonality.

$$
\begin{aligned}
\phi_c(t) &= 2\pi \cdot \frac{t}{\text{day}}, \\
\theta_c(t) &= \sin\left(2\pi \cdot \frac{t}{\text{year}}\right) \cdot \theta_{\max}, \\
F(\phi, \theta, t) &= h_f^0 \cdot \cos(\phi - \phi_c(t)) \cdot \exp\left(-\frac{(\theta - \theta_c(t))^2}{\sigma^2}\right),
\end{aligned}
$$

where $\phi$ and $\theta$ denote longitude and latitude, respectively; $t$ is the simulation time; $h_f^0$ is the forcing amplitude; and $\phi_c(t)$ and $\theta_c(t)$ represent the seasonally shifting longitude and latitude centers of the forcing. $\theta_{\max} = 0.4$ is the maximum latitudinal declination; $\sigma = \pi/2$ controls the latitudinal spread of the forcing; day and year are time scale constants corresponding to daily and annual cycles. This dataset defines a 42-day year to increase simulation complexity. In our setting, we simulate a half-year period (21 days), with snapshots selected every 8 hours, resulting in a total of 63 time steps, with a spatial resolution of $512 \times 256$. A total of 200 trajectories are generated by sampling the forcing amplitude $h_f^0 \sim \mathcal{U}[0.1, 30]$ and the latitudinal spread parameter $\sigma \sim \mathcal{U}[1, 4]$. These trajectories are split into training (60%), validation (20%), and evaluation (20%) sets.

## C TRAINING DETAILS

### C.1 HYPERPARAMETER CHOICE

**Normalization.** Prior to training, we normalize the data following the approach of (Regazzoni et al., 2024). For bounded data, we scale variables, including the parameter $u$, the space coordinate $\xi$, and the state variable $x$, to the range $[-1, 1]$. For unbounded data, we standardize the variables to zero mean and unit variance. The time step $\Delta t$ in Eq. 4 is treated as a hyperparameter during training.

To determine the optimal hyperparameter choices for LDNets in our examples, we automate the hyperparameter search using Bayesian optimization (Biewald, 2020). The range of hyperparameters considered is listed in Table 4. The term "downsample time steps" refers to the number of time steps

sampled from the original dataset. Meanwhile, "$\Delta t$ normalize" is a constant used to scale the time step $\Delta t$ during latent state evolution. It serves as a practical tool to ensure numerical stability and adjust the effective temporal scale of the latent dynamics, rather than representing a physical time unit. The details of further training and optimized hyperparameters are shown in Table 5. When fine-tuning the reconstruction network, we set the number of epochs to 1000 and the learning rate to $10^{-4}$. We also present the training parameters of the LSTM encoder in Table 6.

Table 4: Hyperparameter search range for LDNets training

| | Dataset | | |
| --- | --- | --- | --- |
| | Tsunami modeling | Kolmogorov flow | Atmospheric modeling |
| Downsampled time steps | $20 - 50$ | $20 - 50$ | $25 - 63$ |
| $\Delta t$ | $0.03 - 0.05$ | $0.04$ | $0.04 - 0.05$ |
| Num of latent states | $8 - 20$ | $2 - 15$ | $10 - 100$ |
| Dynamics net | depth | depth | depth |
| | $5 - 10$ | $2 - 15$ | $5 - 10$ |
| | width | width | width |
| | $10 - 200$ | $20 - 200$ | $100 - 200$ |
| Reconstruction net | depth | depth | depth |
| | $5 - 12$ | $2 - 15$ | $8 - 15$ |
| | width | width | width |
| | $100 - 400$ | $20 - 700$ | $300 - 500$ |
| Fourier embedding dim | $0 - 50$ | $0 - 50$ | $10 - 100$ |
| StepLR (gamma) | $0.1 - 0.8$ | $0.1 - 0.9$ | $0.1 - 0.7$ |
| Batch size | $2 - 16$ | $2 - 16$ | $1 - 4$ |

Table 5: Training details for LDNets

| | Dataset | | |
| --- | --- | --- | --- |
| | Tsunami modeling | Kolmogorov flow | Atmospheric modeling |
| Downsample time step | 50 | 40 | 63 |
| $\Delta t$ | 0.036 | 0.04 | 0.04 |
| Space points | 5000 | 5000 | 10000 |
| Dynamics net | MLP | MLP | MLP |
| | 8 hidden layers | 9 hidden layers | 8 hidden layers |
| | 50 hidden dim | 200 hidden dim | 200 hidden dim |
| | ReLU | ReLU | ReLU |
| Reconstruction net | MLP | MLP | MLP |
| | 10 hidden layers | 14 hidden layers | 15 hidden layers |
| | 300 hidden dim | 500 hidden dim | 500 hidden dim |
| | ReLU | ReLU | ReLU |
| Fourier embedding dim | – | 10 | 50 |
| Initialization | Glorot normal | Glorot normal | Glorot normal |
| Adam (lr) | $10^{-3}$ | $10^{-3}$ | $10^{-3}$ |
| StepLR (gamma, step size) | $(0.6, 200)$ | $(0.7, 200)$ | $(0.6, 200)$ |
| Batch size | 2 | 6 | 2 |
| Epoch | 2000 | 2000 | 2000 |
| Loss | MSE | MSE | MSE |

## C.2   OFFLINE COST VS. ONLINE GAIN

Training the LDNets takes 12.08 hours for the Kolmogorov flow, 10.76 hours for tsunami modeling, and 21.86 hours for atmospheric modeling on a single NVIDIA RTX A6000 GPU. The training process can be further accelerated by utilizing multiple GPUs, and further code optimization may improve efficiency and reduce computational costs.

Although LD-EnSF incurs a higher initial training cost, this cost is quickly offset by its significantly lower per-cycle assimilation cost. Based on a practical cost breakdown, LD-EnSF becomes more computationally efficient than Latent-EnSF when the ensemble size exceeds 100 in a single assimilation cycle. To quantify this, we use the following total cost formulation for $n$ assimilation cycles.

Table 6: Training details for LSTM encoder

|  | Dataset | | |
| --- | --- | --- | --- |
|  | Tsunami modeling | Kolmogorov flow | Atmospheric modeling |
| Hidden layers | 1 | 1 | 1 |
| Hidden dim | 256 | 128 | 512 |
| Initialization | Glorot normal | Glorot normal | Glorot normal |
| Adam (lr) | $10^{-3}$ | 0.002 | $10^{-4}$ |
| CosineAnnealingLR (Tmax, eta min) | – | $(5000, 10^{-4})$ | $(5000, 10^{-4})$ |
| Epoch | 20000 | 75000 | 75000 |
| Dropout | 0.1 | 0 | 0 |

($T_f$, $T_r$, and $T_d$ in Table 2 are computed based on 100 trajectories):

$$T_{\text{total}} = T_{\text{data}} + T_{\text{training}} + \frac{n}{100}(T_f + T_r + T_d)$$

Take atmospheric modeling as an example. Combining the data in Table 2 and given that $T_{\text{data}} = 298$ seconds, it is easy to see that when $n > 100$, LD-EnSF becomes more efficient than other methods. Furthermore, there are many scenarios where real-time assimilation is crucial. For example, in tsunami modeling, we can spend time on offline training beforehand; however, once a tsunami occurs, we need predictions as quickly as possible for effective forecasting. In such cases, LD-EnSF is about $4 \times 10^3$ times faster than Latent-EnSF and other methods during assimilation, making it highly suitable for time-sensitive applications.

## D    SMOOTHNESS OF LATENT STATES

A comparison of the VAE and LDNet latent spaces for the Kolmogorov flow and atmospheric modeling is shown in Fig. 6.

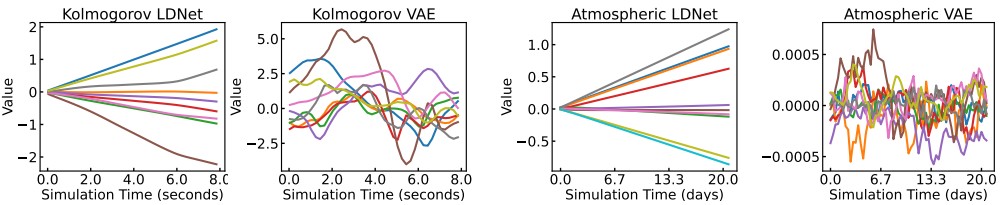

Figure 6: Latent states of Kolmogorov flow and atmospheric modeling.

Due to the smoothness of the LDNet latent space, the output can be evaluated at arbitrary continuous time steps by interpolation. In the tsunami modeling, Fig. 7 reports the reconstruction error based on these interpolated latent states, where the model is trained on 50 time steps sampled from a total of 2000 physical time steps. During testing, the latent trajectories are interpolated to 400 time steps for reconstructing the full physical states.

## E    ROBUSTNESS TO NOISE AND DISTRIBUTION SHIFT

### E.1    ESTIMATING THE UNCERTAINTY OF LATENT STATES

To estimate latent observation noise $\hat{\gamma}_t$, we first train the LSTM encoder using noise-free observations. We then encode noisy observations and compare the resulting latent states to the reference latent states obtained from a trained LDNet. $\hat{\gamma}_t$ is then estimated as the standard deviation of the difference between the noisy and reference latent states. Fig. 8 shows the estimated $\hat{\gamma}_t$, in which we assume a uniform noise level across latent dimensions.

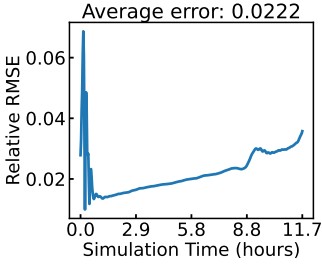

Figure 7: Test error of reconstructing full physical states from interpolated latent states in tsunami modeling.

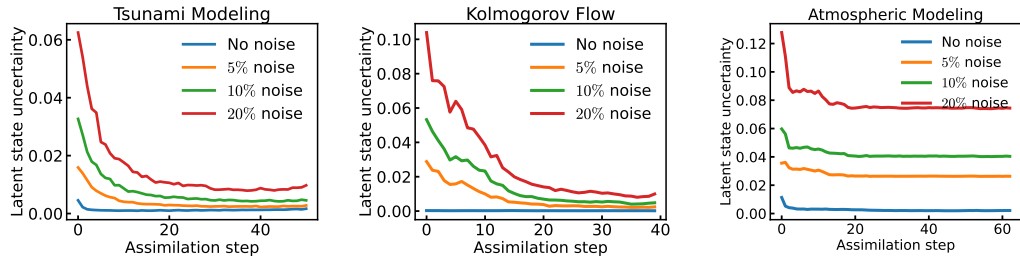

Figure 8: Estimated uncertainty in the latent space, measured as the standard deviation of the difference between the encoded latent states (from noisy observations) and the true latent states obtained from the trained LDNet.

## E.2 Non-Gaussian / Heteroskedastic Noise

To evaluate the robustness of LD-EnSF under more realistic observation conditions, we conduct experiments using several non-Gaussian and heteroskedastic noise models in the tsunami assimilation task. All noise distributions are scaled to achieve a mean noise-to-signal ratio of approximately 10%. Table 7 reports the assimilation error across these settings.

Table 7: Assimilation error under different observation noise models.

| Noise Type | Gaussian | Nonstationary Gaussian | Sinusoidal Heteroskedastic | Beta | Multivariate Gaussian |
|---|---|---|---|---|---|
| Assim. Error | 0.053 | 0.086 | 0.061 | 0.072 | 0.064 |

The **nonstationary Gaussian** setting introduces a time-dependent mean that varies across assimilation steps but is spatially uniform. The **sinusoidal heteroskedastic** setting applies temporally varying Gaussian noise whose variance oscillates between 0% and 20% of the signal amplitude, following a sine waveform. The **Beta noise** is sampled from a scaled Beta distribution (e.g., $\text{Beta}(2, 5)$), inducing asymmetric, bounded perturbations. Finally, the **multivariate Gaussian** model introduces correlated noise with covariance matrix $\Sigma = AA^\top$, where $A$ contains i.i.d. standard normal entries.

Despite the increased variability and structural complexity of these noise models, LD-EnSF consistently achieves low assimilation error, demonstrating robustness beyond the standard isotropic Gaussian setting.

## E.3 Out-of-Distribution Generalization

While out-of-distribution (OOD) generalization is not the main focus of this work, it remains an important consideration for real-world deployment. We evaluate the robustness of LDNet under distribution shift by testing it on trajectories initialized with Gaussian bumps displaced outside the

training region. We evaluate the OOD generalization of LDNet by testing it on trajectories whose initial Gaussian bump lie outside the training region. Specifically, while training conditions are sampled from $[0, 0.5L] \times [0, 0.5L]$, we test on locations such as $[0.25L, 0.52L]$, $[0.25L, 0.54L]$, etc. (Table 8).

Table 8: LDNet test error under increasing displacement from the training region. The horizontal axis indicates the center shift $L$ of the initial Gaussian bump.

| Distance to Train Region | $0.02L$ | $0.04L$ | $0.06L$ | $0.08L$ | $0.10L$ |
|---|---|---|---|---|---|
| LDNet Error | 0.0553 | 0.0720 | 0.0889 | 0.104 | 0.122 |

As expected, the error increases with distance from the training region, indicating a consistent degradation under distribution shift. However, the extent of generalization is problem dependent. For instance, the original LDNet paper (Regazzoni et al., 2024) demonstrates strong extrapolation in time on unsteady Navier–Stokes flows.

### E.4 UNSTRUCTURED OBSERVATIONS ASSIMILATION RESULTS UNDER NOISE

By leveraging the LSTM encoder, LD-EnSF can accommodate unstructured observations at arbitrary locations. However, training a new LSTM is required for different sets of observation points. In the following results, 100 observation points ($0.44\%$) are randomly selected as the observation set. Training the LSTM observation encoder to the observations with latent states using the hyperparameter setting in Table 6 achieves a test relative RMSE of $0.094\%$ for the Kolmogorov flow, $0.83\%$ for the tsunami modeling and $2.68\%$ for atmospheric modeling. The visualizations are in Appendix H.

Under random observation settings, we present the assimilation results in Fig. 9. We conduct 20 experiments, each with a different ground-truth trajectory, and report the mean and uncertainty of our method. The figure also includes results under different noise levels, demonstrating the robustness of LD-EnSF. For the Kolmogorov flow and atmospheric modeling, since the data is normalized to have a mean of 0 and a standard deviation of 1, Gaussian noise is applied with a standard deviation proportional to the specified noise level, e.g., a standard deviation of 0.1 for $10\%$ noise. For the tsunami modeling, due to the long-tailed nature of the data distribution, Gaussian noise is added adaptively based on the value at each observation point.

Note that Fig. 9 (bottom middle) shows the results of parameter estimation, where the purple line corresponds to the no-assimilation baseline and exhibits high variance. It corresponds to the no-assimilation baseline and reflects the prior distribution of the parameters before any data assimilation is applied. The reported variance is computed across multiple test cases with different ground-truth trajectories, and the wide spread of the purple curve reflects the variability in true parameter values across these cases. This highlights the inherent difficulty of parameter estimation in the absence of assimilation, particularly when the parameter is not directly observed and affects the system only indirectly. In contrast, the assimilation results (depicted by the remaining curves) show substantially reduced variance and lower mean errors. For instance, under 20% observation noise, the mean estimation error is approximately 10%, with a standard deviation of around 15%. This performance is comparable across most benchmarks, though atmospheric modeling remains more challenging because of the weak sensitivity of the observed field to the external forcing parameter. This makes the inverse problem more ill-posed, especially under sparse observations. In terms of convergence behavior, the estimated parameters generally stabilize within approximately 10 assimilation steps, which is consistent with other benchmark settings.

### E.5 STRUCTURED OBSERVATIONS ASSIMILATION RESULTS UNDER NOISE

Fig. 10 shows the assimilation performance of LD-EnSF under varying levels of observation noise. Results are averaged over 20 runs with different ground-truth trajectories, and both the mean and uncertainty estimates are reported. Note that the inferred parameter $u$ exhibits higher uncertainty, which is expected, as recovering the system parameters from limited and noisy data is inherently ill-posed.

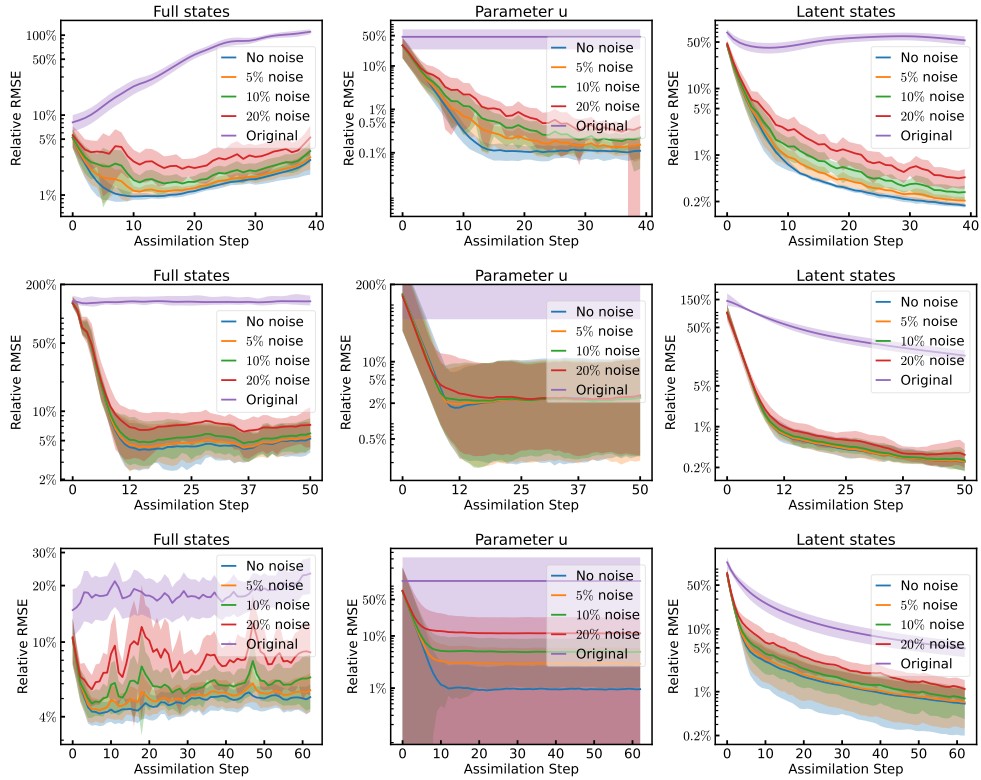

Figure 9: Assimilation results for unstructured observations in the Kolmogorov flow(top), tsunami modeling (middle), and atmospheric modeling (bottom). The left panel shows the relative RMSE of full states, while the middle and right panels display the error of the assimilated parameters and latent states, respectively, compared to the latent states at the true parameters. For reference, errors of the original (unassimilated) quantities are also included.

# F    SENSITIVITY AND STABILITY ANALYSIS

Empirically, we conducted additional experiments that vary the latent dimension, ensemble size, and observation sparsity for the tsunami modeling problem. To facilitate comparison, To facilitate comparison, the assimilation error shown in this section is defined as the relative RMSE at the final time step, while the surrogate model error refers to the relative RMSE averaged over time.

## F.1    LATENT DIMENSION SENSITIVITY

In our tsunami modeling experiments, we investigate how varying the latent dimension influences both the surrogate model accuracy and the assimilation performance, while keeping other hyperparameters unchanged. As shown in Table 9, a clear failure mode emerges when the latent space is restricted to 1 or 2 dimensions. The best performance, in terms of relative RMSE, is achieved with a moderate latent dimension between 5 and 10. While a high-dimensional latent space (e.g., 500 dimensions) does not lead to complete failure, it results in significantly higher errors compared to these more balanced settings.

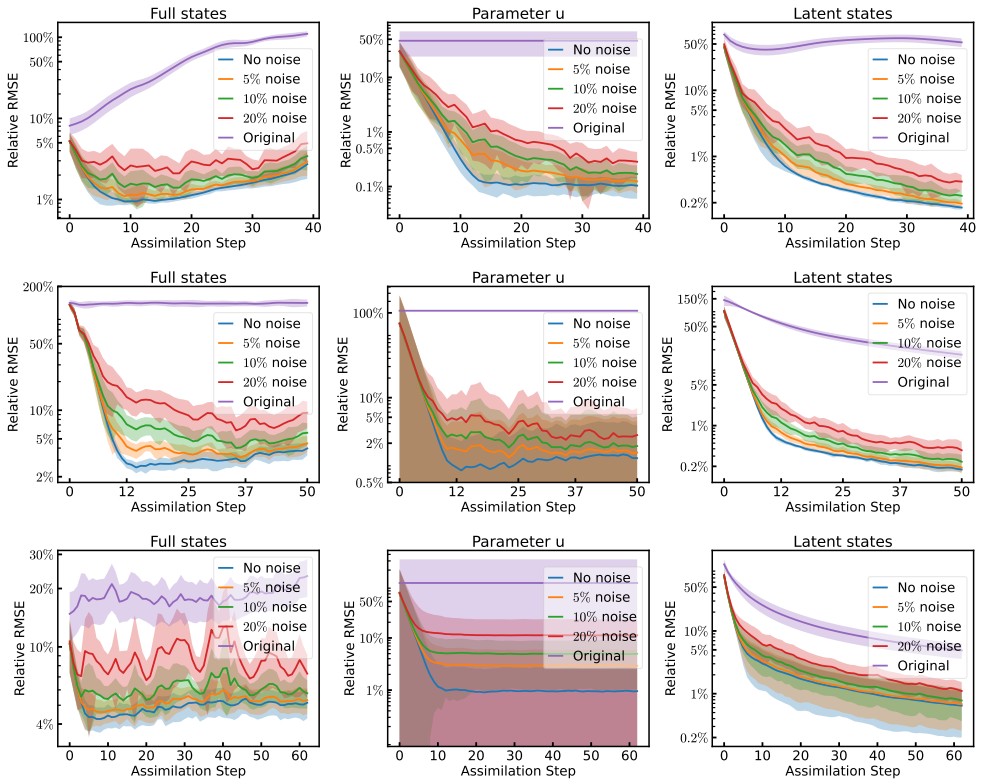

Figure 10: Assimilation results for grid-based observations in the Kolmogorov flow(top), tsunami modeling (middle), and atmospheric modeling (bottom). The left panel shows the relative RMSE of full states, while the middle and right panels display the error of the assimilated parameters and latent states, respectively, compared to the latent states at the true parameters. For reference, errors of the original (unassimilated) quantities are also included.

Table 9: Effect of latent dimension on the LDNet surrogate error (in relative RMSE) and assimilation performance for tsunami modeling.

| Latent Dim | 1 | 2 | 3 | 5 | 10 | 50 | 100 | 200 | 500 |
|---|---|---|---|---|---|---|---|---|---|
| LDNet Error | 0.861 | 0.459 | 0.0225 | 0.0216 | 0.0213 | 0.0239 | 0.0271 | 0.0230 | 0.0473 |
| Assim. Error | 0.869 | 0.387 | 0.0430 | 0.0364 | 0.0371 | 0.0504 | 0.0528 | 0.0474 | 0.0993 |

## F.2 ENSEMBLE SIZE SENSITIVITY

EnSF consists of two components: the prior score estimation (done through Monte-Carlo sampling) and the likelihood score estimation. The latent observation in the likelihood is defined over the entire latent space, whose contribution dominates the construction of the posterior score, a point explored in Latent-EnSF (Section 4.1.1). With one ensemble member, the algorithm effectively reduces to a MAP estimate and yields similar accuracy. For LD-EnSF, increasing the ensemble size has minimal impact on the average error, as shown in Figure 11, which reports both the root-mean-square error (RMSE) and the Continuous Ranked Probability Score (CRPS) Matheson & Winkler (1976). While the mean errors remain stable, the variance across 20 different trajectories increases with ensemble size, reflecting the uncertainty introduced by sparse and noisy observations. Note that a small ensemble size was shown to be sufficient in the latent assimilation, see Figure 5 in (Si & Chen, 2025), while a large ensemble size has to be used for the full/original space assimilation to increase its accuracy.

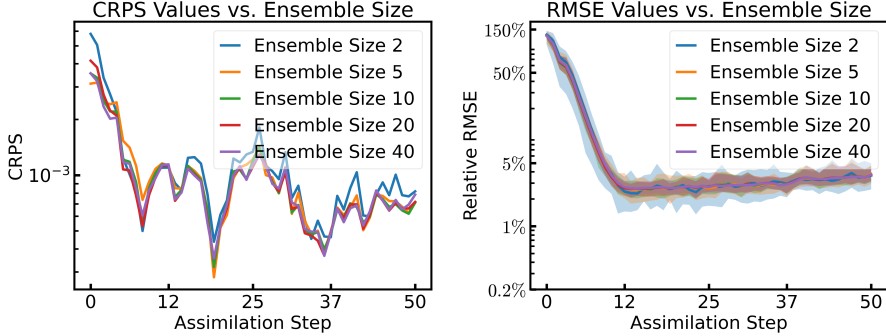

Figure 11: CRPS and RMSE values for varying ensemble size in tsunami modeling

## F.3 Surrogate Model Sensitivity

The accuracy of the surrogate models (LDNet and LSTM encoder) plays a critical role in the overall assimilation performance. In our setting, the LSTM is trained on latent trajectories generated by LDNet, so poor LDNet accuracy also affects the quality of the observation encoder. Therefore, it is essential to ensure that LDNet reaches a reasonable level of fidelity. In Table 10, the assimilation error decreases as LDNet error decreases during training. Notably, even when the LSTM achieves similar test errors across different LDNet models, the assimilation accuracy still depends strongly on the quality of LDNet.

This gap between surrogate and assimilation error is expected. While the LDNet is evaluated using full and accurate latent input, the assimilation process relies on partial and noisy observations, which are encoded by a separately trained LSTM. The LSTM itself introduces approximation error when reconstructing latent trajectories, and during testing we additionally add 10% noise to the observations. Therefore, it is reasonable, and theoretically consistent, that assimilation results exhibit higher error, e.g., 0.0371, than the surrogate model error, e.g., 0.0213 at epoch 1999.

Table 10: Effect of LDNet training duration on surrogate and assimilation accuracy in tsunami modeling.

| Epoch | 99 | 299 | 399 | 499 | 799 | 999 | 1999 |
|---|---|---|---|---|---|---|---|
| LDNet Error | 0.144 | 0.0794 | 0.0483 | 0.0359 | 0.0309 | 0.0256 | 0.0213 |
| LSTM Error | 0.0041 | 0.0033 | 0.0033 | 0.0030 | 0.0029 | 0.0032 | 0.0028 |
| Assim. Error | 0.252 | 0.0975 | 0.0693 | 0.0499 | 0.0443 | 0.0411 | 0.0371 |

## F.4 Observation Density Sensitivity

We also study the impact of observation density on assimilation accuracy. As expected, the assimilation error increases as fewer observations are available, due to the reduced information available for inference. The results are summarized in Table 11.

Table 11: Assimilation error under varying observation density in tsunami modeling.

| Observation Density | 0.08% | 0.10% | 0.16% | 0.25% | 0.44% | 1.00% |
|---|---|---|---|---|---|---|
| Assimilation Error | 0.129 | 0.0891 | 0.0476 | 0.0504 | 0.0371 | 0.0353 |

## F.5 $\gamma$ Sensitivity

To test the sensitivity of LD-EnSF to misspecified $\hat{\gamma}_t$, we varied the scalar value used in latent space. As shown in Fig. 12, using a $\hat{\gamma}_t$ smaller than the estimated value leads to a mild increase in error

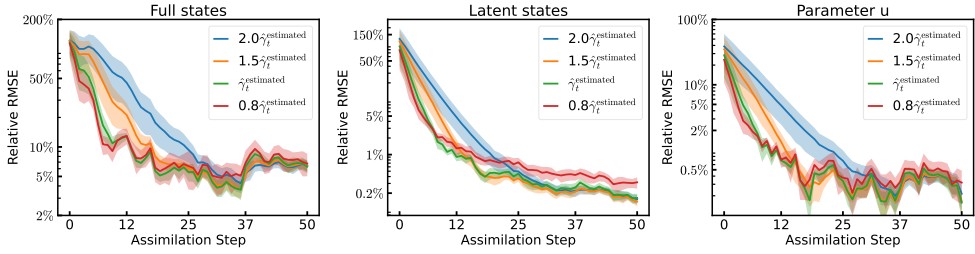

Figure 12: Assimilation results for misspecified $\hat{\gamma}_t$ in Eq. 8

during assimilation. When $\hat{\gamma}_t$ is larger than the estimated value, the initial error is higher, but the assimilation still converges to low error. Overall, LD-EnSF remains stable across a wide range of $\hat{\gamma}_t$ values, indicating that the method is not highly sensitive to moderate misspecification of the latent noise level.

## G  ABLATION STUDY ON KEY COMPONENTS

We evaluate how each component of LDNet, the observation encoder, the training pipeline, and integration for data assimilation contributes to overall performance.

### G.1  LDNET ARCHITECTURE AND TRAINING STRATEGY

Several improvements have been made to LDNet, including the incorporation of Fourier encoding, the use of ResNet blocks, and a fine-tuning strategy. In Table 12, we use the Kolmogorov flow example to isolate and evaluate the individual contributions of these architectural enhancements. We observe that the combination of ResNet and Fourier embedding largely improves the performance of LDNet.

Table 12: LDNet performance on Kolmogorov Flow with different architectural ablations.

| Kolmogorov Flow | w/ Fourier & ResNet | w/ ResNet | w/ Fourier | w/o ResNet/Fourier |
|---|---|---|---|---|
| LDNet Error | 0.0223 | 0.0237 | 0.0268 | 0.0340 |

As shown in Table 13, fine-tuning of the reconstruction network improves accuracy beyond initial training of our proposed LDNet, and both stages significantly outperform the original LDNet.

Table 13: Relative RMSE for different training strategies in tsunami modeling.

|  | Training | Fine-tuning | Original LDNet |
|---|---|---|---|
| Tsunami | 0.0213 | 0.0168 | 0.1837 |
| Kolmogorov | 0.0223 | 0.0123 | 0.0349 |

While some works (Tancik et al., 2020) keep the Fourier feature matrix $B$ fixed by sampling it from a Gaussian distribution, in this paper we follow Salvador & Marsden (2024) and treat $B$ as a trainable parameter. As shown in Fig. 13, whether $B$ is fixed or trainable does not affect the training behavior. This observation is consistent with Appendix A.3 of Tancik et al. (2020), where optimizing the Fourier feature parameters yields nearly identical results. In addition, using a trainable $B$ removes the need to tune the sampling distribution of the fixed random matrix.

### G.2  OBSERVATION ENCODER ARCHITECTURE

We compare our LSTM observation encoder with a CNN-based encoder composed of residual blocks and an intermediate self-attention layer, which is also used by Si & Chen (2025). For tsunami

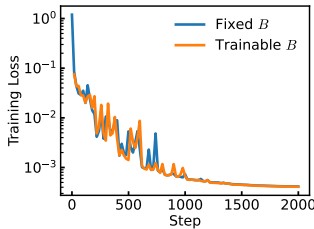 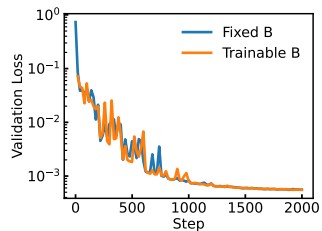

| Setting | Test Error |
|---|---|
| Fixed $B$ | 0.0221 |
| Trainable $B$ | 0.0220 |

Figure 13: Training loss, validation loss, and test error in Kolmogorov with either fixed or trainable $B$ in the Fourier encoding.

modeling, we evaluate four configurations and compare the assimilation error: (1) a CNN encoder with direct full-state reconstruction from encoded observations; (2) LD-EnSF with a CNN encoder; (3) an LSTM encoder alone; and (4) LD-EnSF with an LSTM encoder. The LSTM encoder, which incorporates temporal context, achieves significantly lower latent representation error (0.5% vs. 0.98%) than the CNN encoder. As shown in Fig. 14, this also translates to significantly improved assimilation accuracy.

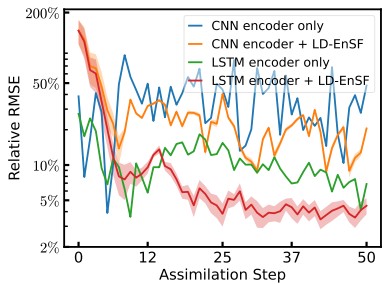

Figure 14: Comparing observation encoders with and without filtering. The colored regions indicate estimated uncertainties for observation noise at a $10\%$ level.

### G.3  Two Phase Training of LDNet and Observation Encoder

Although LDNet can, in principle, be trained jointly with the LSTM encoder to simplify the workflow, we find that balancing the different loss terms is nontrivial and does not lead to improved performance. To evaluate this approach, we experimented with a combined training objective:

$$\mathcal{L}_{\text{total}} = \mathcal{L}_{\text{latent}} + \mathcal{L}_{\text{LSTM}} + \mathcal{L}_{\text{obs}},$$

where the first two terms are defined in Eqs. 5 and 7, and $\mathcal{L}_{\text{obs}}$ measures the reconstruction error from LSTM-predicted latents. This joint training strategy results in a surrogate model error of 0.0241 and an assimilation error of 0.0693 for tsunami modeling, which are higher than the results obtained using separate training in Table 1.

### G.4  With and Without Latent-Space Assimilation

In Fig. 15, we compare the reconstruction accuracy of two approaches for tsunami modeling under different noise levels: (i) directly reconstructing the full state from latent observations encoded by the LSTM encoder (denoted as LSTM-only, without any filtering), and (ii) LD-EnSF, which performs ensemble-based filtering in the latent space. The LSTM-only baseline performs well when there is no observation noise. However, as noise increases (e.g., at 5% and 10%), its performance degrades significantly. In contrast, LD-EnSF maintains a lower reconstruction error by effectively incorporating both prediction uncertainty and observation uncertainty through latent-space filtering. Moreover, LD-EnSF is able to estimate output uncertainty using the ensemble, while directly reconstructing the

full state using the LSTM encoder and reconstruction network cannot. Reconstruction from arbitrarily located sparse sensors has been studied independently by Williams et al. (2024); Tong & Chen (2026). This highlights the benefit of applying the EnSF on top of the LSTM encoder, especially under noisy conditions.

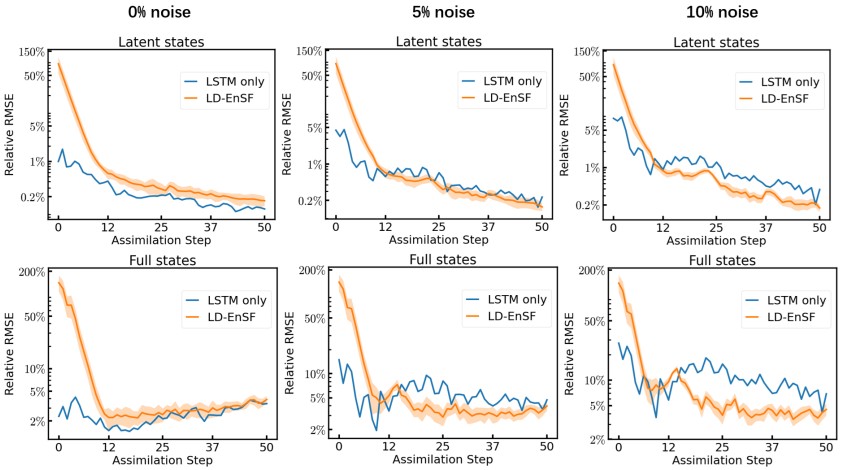

Figure 15: Comparison between the LSTM-only baseline and LD-EnSF assimilation results on tsunami modeling with 0%, 5%, and 10% observation noise. The colored regions indicate uncertainty estimated by the ensemble.

## H ASSIMILATION RESULTS VISUALIZATION

This section visualizes the assimilation process, including observation points, reconstructed states, and assimilation errors. Figs. 17 and 16 show results for tsunami modeling and the Kolmogorov flow, respectively, comparing structured ($10 \times 10$ grid) and randomly sampled (100 points) observations. In Fig. 17, although the trajectory starts from a misspecified initial condition, assimilation corrects it over time. In Fig. 16, the trajectory with an incorrect $Re$ deteriorates rapidly, while assimilation with LD-EnSF effectively reduces the error. The assimilated dynamics for atmospheric modeling are shown in Fig. 18. A comparison of different data assimilation methods at the final time step, in terms of meridional wind and geopotential height, is presented in Fig. 19.

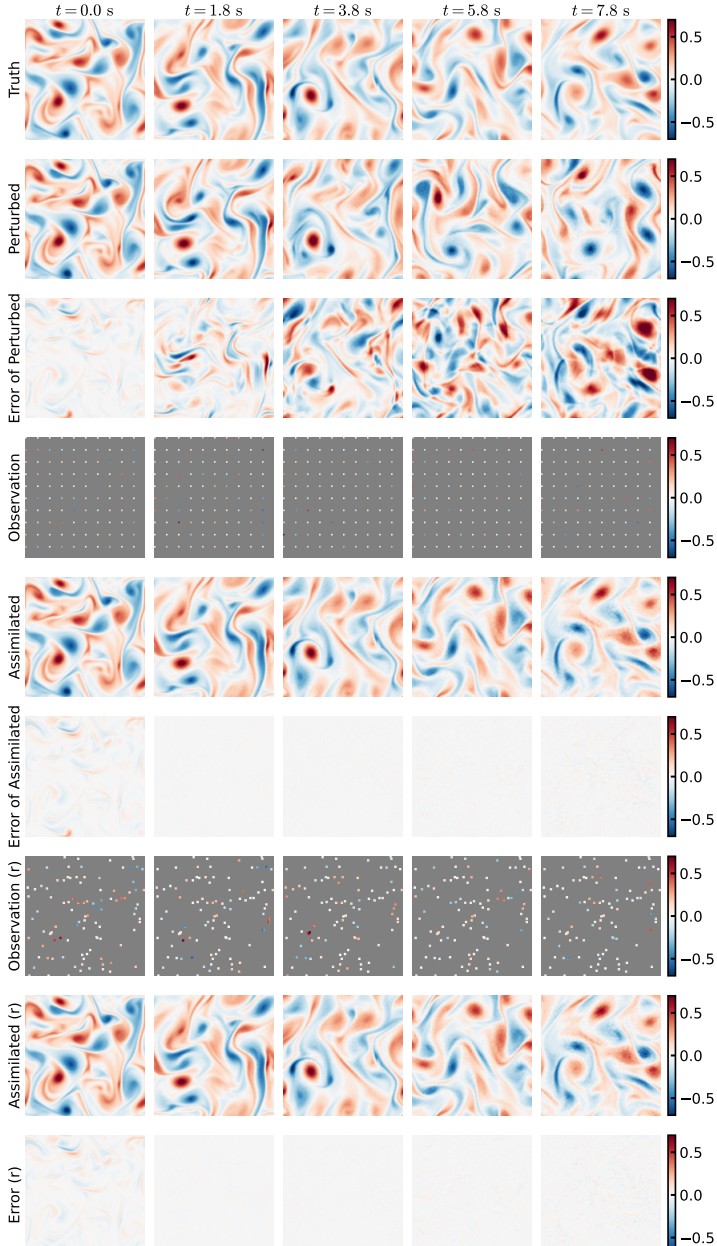

Figure 16: Visualization of the vorticity field $\omega = \nabla \times \mathbf{v}$ of the ground truth of the Kolmogorov flow in at a Reynolds number of $Re = 674.37$ (1st row) and the perturbed dynamics at $Re = 1469.5$ (2nd row), with their difference shown in the 3rd row. Sparse observations ($10 \times 10$ from $150 \times 150$ grid) (4th row) are assimilated into an ensemble of 20 LDNet trajectories via LD-EnSF. The 5th row shows one trajectory from the ensemble, starting from a deviated $Re = 1469.5$. The 6th row presents assimilation errors. The last three rows correspond to 100 randomly sampled observation points and their respective assimilation results, where "r" denotes random.

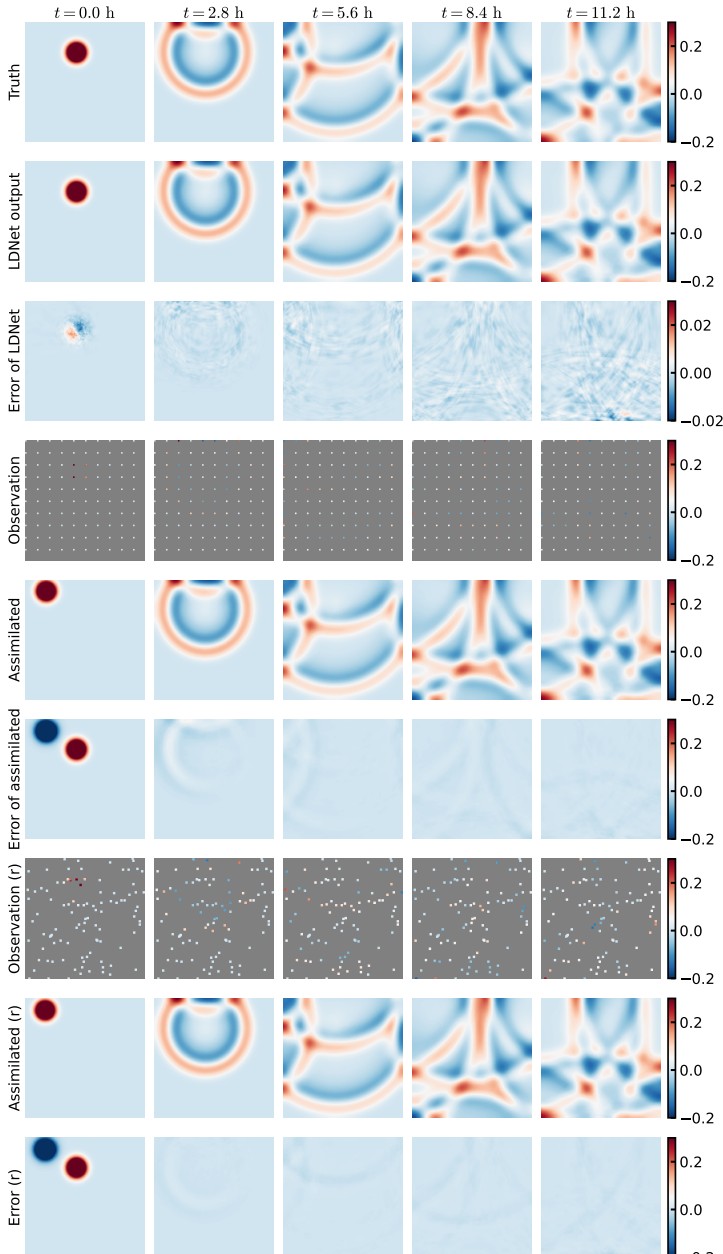

Figure 17: Visualization of the surface elevation $\eta$ in tsunami dynamics: (1st row) ground truth, (2nd row) LDNet predictions from known initial conditions, and (3rd row) prediction errors. Sparse observations ($10 \times 10$ from $150 \times 150$ grid) (4th row) are assimilated into an ensemble of 20 LDNet trajectories via LD-EnSF. The 5th row shows one trajectory from the ensemble, starting from a deviated initial condition. The last three rows correspond to 100 randomly sampled observation points and their respective assimilation results, where 'r' denotes random.

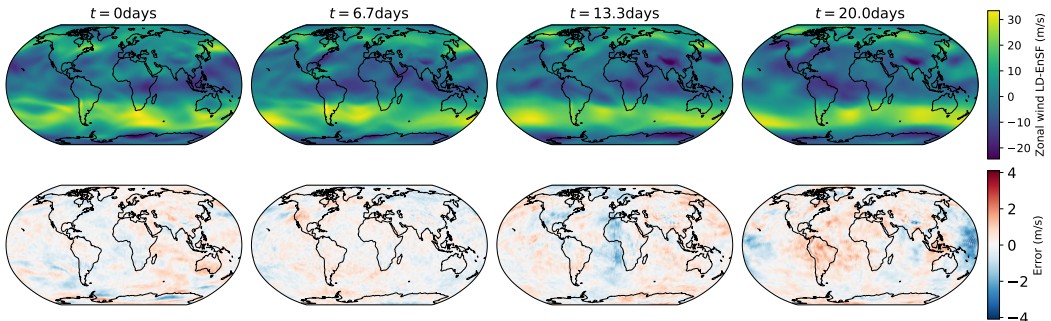

Figure 18: Visualization of LD-EnSF assimilated dynamics over 21 days. The bottom row shows the corresponding assimilation errors.

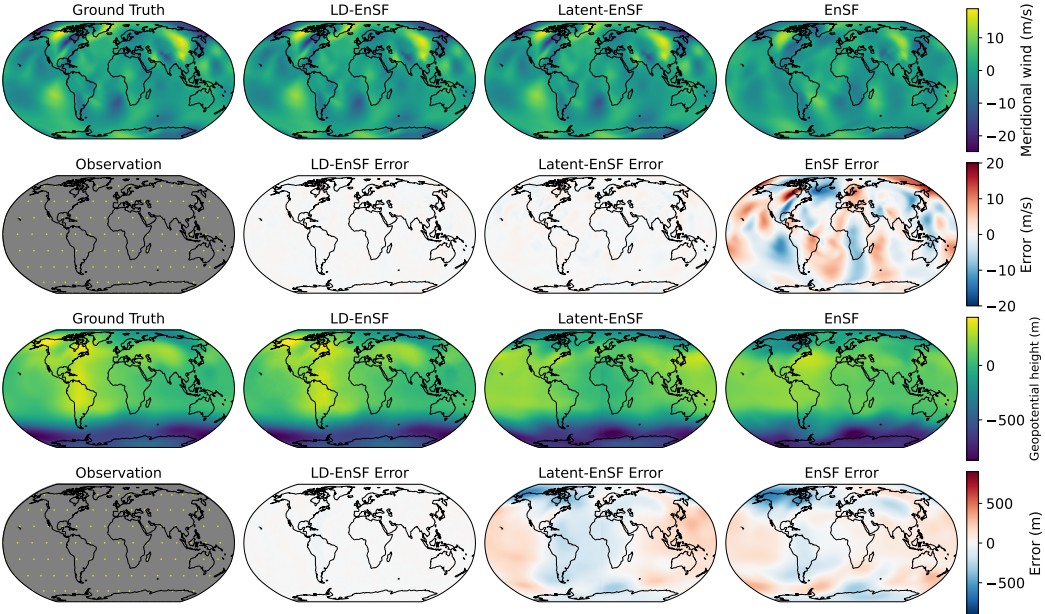

Figure 19: Comparison of LD-EnSF, Latent-EnSF, and EnSF in terms of meridional wind and geopotential height at the final assimilation step.

