# OpenReview forum: "LD-EnSF: Synergizing Latent Dynamics with Ensemble Score Filters for Fast Data Assimilation with Sparse Observations"
_ICLR.cc/2026/Conference — ICLR 2026 Poster_

### Official Review · Reviewer_YcmD · 2025-10-26

**Soundness:** 1
**Presentation:** 2
**Contribution:** 1
**Rating:** 2
**Confidence:** 5

**Summary:**

## Summary
This paper introduces LD-EnSF, a data-driven framework for data assimilation (DA) that aims to solve two major challenges: the high computational cost of numerical simulations and the difficulty of applying score-based methods to sparse observations. The core idea is to (1) learn a fast surrogate model that forwards the system dynamics in a low-dimensional latent space, and (2) use a LSTM encoder to map sparse observations into this latent space, where EnSF is then performed. The numerical experiments on three challenging high-dimensional benchmarks show that LD-EnSF achieves notable speedups while maintaining high accuracy even with sparse observations.

**Strengths:**

## Strengths
- The authors provide a comprehensive numerical studies on three challenging, high-dimensional benchmarks and present solid comparisons of LD-EnSF against strong baselines. The results show the advantages of proposed methods in computational efficiency and assimilation accuracies.

**Weaknesses:**

## Weakness
- The core idea of performing EnSF in a latent space was already achieved by Latent-EnSF [Si and Peng, 2025]. This paper's primary contributions are (1) replacing the VAE encoder from Latent-EnSF with an LSTM encoder and (2) replacing the full-space dynamics with an LDNet surrogate. While this is a successful *engineering* effort that yields speedups and imporved performances, the novlties in machine learning for DA is minimal. The paper does not provide sufficient theoretical justification for why this specific combination is fundamentally superior, beyond empirical performance.
- The paper claims the latent representation addresses sparsity. However, the actual mechanism enabling the system to work is the "*history-aware* LSTM encoder". By processing a sequence of sparse observations $(y_{1:t})$, the LSTM is implicitly learning a time-delay embedding. It is a well-established concept (e.g., via Takens' embedding theorem [Takens et al. 1981; Noakes et al., 1991]). The paper does not discuss this mechanism entirely. And, It lacks a large and highly relevant recent advances of such works for DA, for example Gottwald et al., 2021; Tarumi et al, 2025; Yang et al, 2025.
- Several key statements and claims regarding the method's motivation and mechanism are vague and require further clarification and justification, see Questions.

**Questions:**

## Questions

- In Line 47, what is meant by *score becomes ill-posed*? This phrasing is imprecise. Could you clarify? The term *ill-posed* refers to problems (typically inverse problems) that fail to meet one or more of the following criteria: a solution exists, the solution is unique, and the solution depends continuously on the data. There is no such definition for a function itself, e.g., score function. Do you mean the estimation of the score from sparse data is high-variance, or simply that the likelihood gradient is zero in unobserved dimensions?
- The authors claim the latent representation itself addresses observation sparsity, e.g., Line 48-50. But no detailed justification. Could authors elabroate on: (1) How can the latent projection itself solve the sparsity problem, given that the input observation (y_t) is indeed sparse? A latent projection does not create information for that is not already present. (2) Should this capability not be attributed almost entirely to the " LSTM Encoder"? By processing a sequence of sparse observations, this encoder is implicitly learning a time-delay embedding. This concept has been used in data assimilation, yet this is not discussed and cited.
- In Line 50, the authors state that "*latent representations enable more informative gradients*". Could you please explain what "more informative gradients" means in this context? How exactly does working in the latent space make the gradients better for handling sparse observations compared to working in the original space?
- It is a valuable strength to handle the irregular grid observation data which is commen in DA. Could the author include a numerical case study demonstrating the method’s performance on irregular grids?
- In Line 210, the author state the random projection $B$ is trianbale. However, in Random Fourier Features/positional encodings, $B$ is typically fixed [Tancik et al., 2020]; optimizing it can change the random feature distribution across update steps and making the training unstable. This is an unconventional choice and its benefits are not inituitive. Could authors provide justify the reason why set $B$ as trainable and provide an ablation study: fixed versus trainable $B$, including training loss curves and DA accuracy.


## References
- Takens, Floris. "Detecting strange attractors in turbulence." Lecture Notes in Mathematics, Berlin Springer Verlag 898 (1981): 366.
- Noakes, Lyle. "The Takens embedding theorem." International Journal of Bifurcation and Chaos 1.04 (1991): 867-872.
- Gottwald, Georg A., and Sebastian Reich. "Combining machine learning and data assimilation to forecast dynamical systems from noisy partial observations." Chaos: An Interdisciplinary Journal of Nonlinear Science 31.10 (2021).
- Tancik, Matthew, et al. "Fourier features let networks learn high frequency functions in low dimensional domains." Advances in neural information processing systems 33 (2020): 7537-7547.
- Si, Phillip, and Peng Chen. "Latent-EnSF: A Latent Ensemble Score Filter for High-Dimensional Data Assimilation with Sparse Observation Data." The Thirteenth International Conference on Learning Representations (2025).
- Tarumi, Yuta, Keisuke Fukuda, and Shin-ichi Maeda. "Deep Bayesian Filter for Bayes-Faithful Data Assimilation." Forty-second International Conference on Machine Learning (2025).
- Yang, Yiming, et al. "Tensor-Var: Efficient Four-Dimensional Variational Data Assimilation." Forty-second International Conference on Machine Learning (2025).

---

> ### Author Response · Authors · 2025-11-21
> **Rebuttal Reviewer YcmD (Part 1 of 2)**
>
> We thank the reviewer for the constructive suggestions which helped us to improve our paper! Below we address each point with supporting results.
>
> **Contexualizing the Paper**
> > The core idea of performing EnSF in a latent space was already achieved by Latent-EnSF [Si and Peng, 2025]...The paper does not provide sufficient theoretical justification for why this specific combination is fundamentally superior, beyond empirical performance.
>
> A key distinction from Latent-EnSF is that LD-EnSF uses a dynamics-consistent latent space learned by LDNet, whereas the VAE latent in Latent-EnSF does not obey a meaningful temporal evolution and introduces drift during filtering. Second, Latent-EnSF requires decoding the full state at every assimilation step, causing high computational cost and feedback of reconstruction noise; LD-EnSF performs the entire update in latent space and decodes only once, enabling regimes that Latent-EnSF cannot handle efficiently. Finally, LD-EnSF naturally supports parameterized dynamical systems because the latent dynamics are conditioned on $u$, whereas Latent-EnSF has no mechanism to represent parametric variations. These differences are structural rather than incremental and are essential for the sparse, high-dimensional settings studied in the paper.
>
> In terms of theory, to our knowledge, a full convergence theory for latent space data assimilation with learned neural surrogates is till largely open. Existing works provide only partial guarantees. For example, [Cheng et al. (2023)] derives an upper bound on the difference between the surrogate and original latent-space assimilation cost functions. Recent method such as [Mücke et al. (2024)] and [Si et al. (2025)] focus on empirical convergence and scalability. [Li et al (2024)] provide theoretical guarantees only for the surrogate representation, rather than for the latent assimilation procedure itself. While we are unable to give a full convergence analysis of LD-EnSF, we include several empirical assessments in Appendix F, including sensitivity to latent dimension, ensemble size, surrogate model, and observation density.
>
> **Score and Latent Representations in Sparsity**
>
> > In Line 47, what is meant by score becomes ill-posed?
> > In Line 50, the authors state that "latent representations enable more informative gradients". Could you please explain what "more informative gradients" means in this context? How exactly does working in the latent space make the gradients better for handling sparse observations compared to working in the original space?
>
> By “score becomes ill-posed,” we refer to the fact that the gradient of the log-likelihood vanishes in all unobserved dimensions when EnSF is applied directly in the original state space under sparse observations. As shown in Latent-EnSF [Si and Peng, 2025] (Section 2.3), when the observation operator selects only a small subset $S \subset \{1,\ldots,d\}$, the likelihood term $\nabla_x \log P(y_t \mid x_t)$ is zero outside of $S$. This leads to an ill-posed update: in the high-dimensional unobserved region, the filter has no likelihood information to guide the correction, and the score cannot meaningfully update the state. We have added some explanation in Line 50.
>
> Working in a latent space addresses this issue. The encoder maps sparse observations into a dense latent representation, where all latent dimensions receive nonzero gradients. This is what we mean by “more informative gradients”: the likelihood signal, which is sparse or zero in the original space, becomes distributed across the latent dimensions, enabling effective assimilation even when only a small fraction of the physical state is observed. This mechanism directly resolves the vanishing-gradient limitation of EnSF in the original state space and is consistent with the motivation discussed in Latent-EnSF.

---

> ### Author Response · Authors · 2025-11-21
> **Rebuttal Reviewer YcmD (Part 2 of 2)**
>
> **LSTM Encoder**
>
> > Could authors elabroate on: (1) How can the latent projection itself solve the sparsity problem, given that the input observation (y_t) is indeed sparse?... (2) Should this capability not be attributed almost entirely to the " LSTM Encoder"?
>
> The LSTM encoder does provide temporal aggregation and effectively creates a time-delay embedding, and we have added the corresponding citation at Line 222. However, this is not the main reason sparse data assimilation becomes possible. As shown in Latent-EnSF, even a VAE-style encoder,without temporal memory, can support sparse assimilation. The key mechanism is the mapping from sparse observations into a structured latent space, which allows the model to extrapolate based on patterns learned during training.
>
> A data-driven latent space also implicitly learns the background prior covariance, whereas traditional DA methods must specify these covariances manually. As illustrated in [Fan et al. (2025)], the encoded latent covariance is much closer to diagonal compared to the physical state space, making EnSF updates more effective. The role of the LSTM is to provide a less noisy latent observational representation than the VAE encoder, particularly by using temporal context. However, the LSTM alone does not incorporate prior estimation. Appendix G.4 shows that when observations are noisy, using only the LSTM encoder leads to degraded performance, whereas performing EnSF assimilation on the latent representations remains robust.
>
> **Citations**
>
> >It lacks a large and highly relevant recent advances of such works for DA, for example Gottwald et al., 2021; Tarumi et al, 2025; Yang et al, 2025.
>
> We thank the reviewer for some additional papers to better position our approach. We have added these citations and discussions referencing the different methods in Related Works.
>
>
> **Irregular Grid Observation Data**
>
> > Could the author include a numerical case study demonstrating the method’s performance on irregular grids?
>
> LD-EnSF takes in observations of irregular grid shapes. In fact, we did a cross comparison on the effects of the grid-based observations (Fig. 10) versus irregular observations (Fig. 9) in the Appendices E.4 and E.5. We also presented the visualization of the irregular observation points in Figs. 16 and 17 in Appendix H.
>
> **Fourier Encoding**
>
> > Could authors provide justify the reason why set B as trainable and provide an ablation study: fixed versus trainable $B$, including training loss curves and DA accuracy.
>
> While Random Fourier Features are often used with a fixed projection matrix B [Tancik et al., 2020], recent work such as [Salvador et al., 2024] treats $B$ as trainable without observing instability. Following this line, we set $B$ to be trainable in LDNet. To address the reviewer’s concern, we have added an ablation study in Appendix G.1 (Fig. 13), comparing fixed and trainable $B$ on the tsunami benchmark. As shown in Fig. 13, training loss, validation loss, and test error are very close in the two settings, and we did not observe any signs of instability. This matches the observation in Appendix A.3 of  [Tancik et al., 2020], where optimizing the Fourier feature parameters yields almost the same performance as keeping them fixed. In addition, learning $B$ removes the need to tune the sampling distribution of the random matrix. We have added this clarification and the ablation figure in the revised version.
>
> **Reference**
>
> [1] Fan, H., Fei, B., Gentine, P., Xiao, Y., Chen, K., Liu, Y., ... & Bai, L. (2025). Physically consistent global atmospheric data assimilation with machine learning in a latent space. arXiv preprint arXiv:2502.02884.
>
> [2] Tancik, Matthew, et al. "Fourier features let networks learn high frequency functions in low dimensional domains." Advances in neural information processing systems 33 (2020): 7537-7547.
>
> [3] Salvador, Matteo, and Alison L. Marsden. "Liquid Fourier Latent Dynamics Networks for fast GPU-based numerical simulations in computational cardiology." arXiv preprint arXiv:2408.09818 (2024).
>
> [4] Cheng, S., Chen, J., Anastasiou, C., Angeli, P., Matar, O. K., Guo, Y.-K., Pain, C. C. & Arcucci, R. (2023). Generalised Latent Assimilation in Heterogeneous Reduced Spaces with Machine Learning Surrogate Models. Journal of Scientific Computing, 94: 11. https://doi.org/10.1007/s10915-022-02059-4
>
> [5] Mücke, Nikolaj T., Sander M. Bohté, and Cornelis W. Oosterlee. "The deep latent space particle filter for real-time data assimilation with uncertainty quantification." Scientific Reports 14, no. 1 (2024): 19447.
>
> [6] Si, Phillip, and Peng Chen. "Latent-EnSF: A Latent Ensemble Score Filter for High-Dimensional Data Assimilation with Sparse Observation Data." The Thirteenth International Conference on Learning Representations (2025).
>
> [7] Li, Zhuoyuan, Bin Dong, and Pingwen Zhang. "Latent assimilation with implicit neural representations for unknown dynamics." Journal of Computational Physics 506 (2024): 112953.

---

> > ### Comment · Reviewer_YcmD · 2025-11-27
> >
> > I thank the authors for their efforts in the rebuttal and the revised manuscript. My point-by-point response is below.
> > - **Contextualizing the Paper**: The authors have clarified the differences from existing works and updated the manuscript accordingly. On theoretical justification, I appreciate the relevant references. I did not intend to require formal analysis for learning latent space with neural network architectures, which would be extremely difficult and not essential here. Instead, I wanted more intuitive theoretical explanations (e.g., via Takens’ embedding theorem) to connect with the mechanism of LSTMs. I am satisfied with the current citations and changes.
> > - **Score and Latent Representations in Sparsity**: The clarification of “ill-posed” and the revised wording are appropriate. I now better understand the point about “informative gradients.” I would phrase it more cautiously: latent encoding can regularize gradients and mitigate vanishing, but does not automatically guarantee informativeness. It may help to state a mild assumption that the physical states lie on a low-dimensional manifold (no need for a formal statement); otherwise a purely injective encoder into a subspace can make information loss and identifiability issues. Overall, I am happy with the revisions on this aspect.
> > - **LSTM Encoder**: The revised explanation of the roles of the LSTM encoder versus the latent EnSF update is helpful. I agree that (i) temporal aggregation via LSTM (time-delay embedding) improves the latent observation quality, and (ii) sparse assimilation fundamentally relies on mapping sparse observations into a structured latent space with a learned prior covariance. The additional experiments (including the “LSTM-only” case in Appendix G.4) and discussion now make this division of roles much clearer, and I find the current positioning reasonable.
> > - **Irregular Grid Observation Data**: Could the authors still clarify what is meant by “irregular grid”? If observations are obtained by random subsampling from a fixed 256×256 grid, they remain grid-aligned. What I had in mind as “irregular” is when observation locations do not lie on the training grid at all (e.g., many observation points fall between training grid points). A brief clarification of this terminology in the manuscript would be useful.
> > - **Fourier Encoding**: The addition of Figure 13 and the related citations adequately addresses my concern.
> >
> > Overall, I am satisfied with the changes and clarifications. I have accordingly increased my scores.

---

> > > ### Author Response · Authors · 2025-11-28
> > >
> > > Thank you very much for the detailed follow up and for reconsidering your scores! We appreciate the constructive feedback and are glad that the revisions clarified your questions. Regarding the terminology on “irregular grid,” we will revise the manuscript to explicitly distinguish between (i) random subsampling from a fixed regular grid (the setting used in our experiments) and (ii) truly irregular observation locations that do not align with training grid points. In our experiment, we adopted the setting (i). Setting (ii) can in principle be handled within our framework, since the LSTM encoder only requires the observation values themselves. We will clarify this distinction in the revised text.
> > >
> > > We sincerely thank the reviewer again for the thoughtful comments!

---

### Official Review · Reviewer_TvkW · 2025-10-29

**Soundness:** 3
**Presentation:** 2
**Contribution:** 2
**Rating:** 4
**Confidence:** 4

**Summary:**

This paper proposes LD-EnSF (Latent Dynamics Ensemble Score Filter), a novel data assimilation method designed for high-dimensional dynamical systems with extremely sparse and irregular observations. The core idea is to avoid expensive full-dimensional simulations during assimilation by operating entirely in a learned latent space. The authors train an improved Latent Dynamics Network (LDNet) to serve as a surrogate model of the system’s dynamics, and a history-aware LSTM encoder to map past sparse observations into the latent space. Assimilation is then performed via the Ensemble Score Filter (EnSF) in the latent space, jointly updating the latent state and system parameters. The method yields orders-of-magnitude speedups (up to $10^5$–$10^6\times$ faster in the experiments) compared to traditional approaches, while maintaining high accuracy and robustness. Extensive experiments on three challenging scenarios (Kolmogorov flow, tsunami propagation, and an atmospheric model) demonstrate that LD-EnSF consistently achieves the lowest estimation errors among strong baselines (LETKF, EnSF, Latent-EnSF), and remains stable even under extreme sparsity where other methods break down. The contributions include: (1) enhancing LDNet with a novel initialization, two-stage training, and a ResNet+Fourier architecture for accurate low-dimensional dynamics; (2) an LSTM-based observation encoder for irregularly-sampled, time-sequential observations; (3) the integration of these components into a fast latent-space score-based filtering framework, delivering real-time capable performance without sacrificing accuracy.

**Strengths:**

Significant Practical Advance: The paper addresses a critical bottleneck in data assimilation – the computational cost under high-dimensional, sparse observation settings. By eliminating full-state simulations during filtering, LD-EnSF achieves massive speedups (e.g. 200,000× in one case), enabling applications (real-time forecasting, larger ensembles) that were previously infeasible. This practical improvement is highly valuable for the community.

Robust Accuracy under Extreme Conditions: Empirical results show state-of-the-art accuracy and robustness. LD-EnSF outperforms LETKF, EnSF, and the prior Latent-EnSF by a clear margin in all tested scenarios. Notably, in an extremely sparse observation scenario (0.1% spatial and 0.2% temporal coverage with 10% noise), LD-EnSF attains about 5% RMSE, whereas LETKF and EnSF either diverge or fail to converge. The method also successfully estimates system parameters (e.g. Reynolds number, forcing amplitude) alongside the state, which is a strong plus.

Well-Motivated Method Design: The integration of learned latent dynamics with score-based filtering is novel and well-justified. The authors identify the shortcomings of VAE-based latent filters (oscillatory latent trajectories, need to propagate in full space) and introduce targeted improvements. In particular, the improved LDNet architecture (with shifted initial latent state handling, two-stage training, and Fourier-featured ResNet decoder) yields a remarkably compact and smooth latent representation that outperforms both the original LDNet and a VAE+LSTM baseline in surrogate modeling accuracy. Similarly, the LSTM-based observation encoder is a sensible choice to capture temporal context and irregular spatial sampling, surpassing the static VAE encoder used in Latent-EnSF. Each design choice is supported by discussion or ablation (e.g. Table 1, Fig. 3 for LDNet vs VAE).

Thorough Experiments: The evaluation is comprehensive. The paper tests three diverse and complex systems, includes strong baseline comparisons (including a “Latent-EnSF-dyn” variant using a VAE-dynamics baseline), and reports detailed metrics. The authors also explore scenarios with varying observation noise (Appendix E.5) and demonstrate the method’s insensitivity to ensemble size (even a single-sample “ensemble” recovers reasonable estimates, akin to MAP). Such thoroughness increases confidence in the results.

Reproducibility: The inclusion of references to code for hyperparameter search (Weights & Biases) and discussion of computational setup (CPU/GPU times in Table 2) is appreciated.

**Weaknesses:**

Dependence on Offline Training and Generalization Limits:
A potential concern is the heavy reliance on training a surrogate model (LDNet) on simulation data before deployment. Acquiring a comprehensive training dataset covering all relevant system behaviors and parameter ranges can be costly, and the method’s performance may degrade if the true system behavior deviates from the training distribution. The paper’s approach is essentially as good as its learned model – for scenarios with significant model uncertainty or evolving dynamics, one might need to frequently retrain or fine-tune the surrogate. This limitation is hinted at in the conclusion (need to retrain adaptively for long-term complex systems), but it remains a practical caveat: the method inherits the generalization limitations of data-driven surrogates.

Incremental Novelty and Comparison with Variational Surrogate Approaches:
While combining latent dynamics learning with score-based filtering is a creative and well-engineered contribution, the methodological novelty of LD-EnSF is largely incremental rather than foundational. The core components—EnKF/EnSF-style ensemble filtering, neural surrogate modeling (LDNet), and LSTM-based latent encoders—are all built upon established techniques. The innovation primarily lies in the system-level integration and practical realization of these elements, rather than in introducing fundamentally new theoretical insights or learning paradigms. Consequently, LD-EnSF may be viewed as an effective and elegant evolution of Latent-EnSF, replacing the VAE and full-state propagation with a better learned surrogate while maintaining the same overall Bayesian filtering framework.

Moreover, when comparing LD-EnSF against variational methods such as 3D-Var or 4D-Var, it should be noted that those methods can also leverage surrogate models like LDNet to reduce computational cost, provided the surrogate is differentiable for adjoint-based optimization. Thus, the runtime advantage of LD-EnSF does not arise solely from operating in latent space, but also depends on whether comparable surrogates are incorporated into the variational baseline. Explicitly discussing this relationship would clarify that LD-EnSF’s main strength lies in its practical integration of latent surrogates with score-based filtering—an important step forward, but not a fundamentally new formulation of data assimilation.

Relation to recent latent-space Bayesian filters:
The paper would benefit from discussing its relationship to other recent approaches that perform Bayesian filtering directly in learned latent spaces, such as the Deep Bayesian Filter (DBF, ICML 2025).
While both LD-EnSF and DBF share the goal of combining learned latent dynamics with probabilistic filtering, they differ in philosophy: DBF integrates inference and dynamics in an end-to-end generative framework, whereas LD-EnSF decouples the surrogate dynamics learning (LDNet) from score-based filtering.
Positioning LD-EnSF more explicitly within this broader family of latent-space Bayesian filters—highlighting differences in training objectives, inference mechanisms, and scalability—would clarify its contribution and increase its relevance to ongoing developments in this research area.

Computational Complexity and Implementation Practicality:
While the paper convincingly demonstrates that LD-EnSF achieves major runtime savings by performing data assimilation in a low-dimensional latent space, its computational efficiency still involves important trade-offs. Each assimilation cycle requires iterative reverse-time SDE integration for $N_e$ ensemble members over $T_{diff}$ diffusion steps, leading to a total cost of $O(N_e T_{diff} d_s)$. This iteration structure is conceptually analogous to the optimization loops in 3D-Var, with the distinction that LD-EnSF's iterations are fully parallelizable across ensemble members but sequential in diffusion time. Moreover, the ensemble size $N_e$ generally needs to grow with the latent dimension to maintain statistical accuracy, as analyzed in recent works (e.g., Oko et al., 2023). Discussing these trade-offs explicitly would help clarify when LD-EnSF offers practical computational advantages over variational baselines, especially since 3D-Var can also benefit from surrogate models such as LDNet if differentiable.

In terms of implementation, the overall pipeline remains quite complex: it integrates multiple neural components (latent dynamics network, reconstruction network, and LSTM-based observation encoder) with a non-trivial score-based assimilation algorithm. Implementing and tuning such a system—including hyperparameter searches, ensuring stable LDNet training, and discretizing the SDE solver—requires substantial expertise. While the methodology is sound, this complexity may pose barriers to adoption compared to simpler ensemble or variational filters. Providing open-source code or detailed pseudocode would greatly enhance reproducibility and accessibility for the community.

Limitation and Clarity of Latent-State Initialization:
The paper initializes all latent trajectories with a fixed zero state ($s_{-1} = 0 $) and relies on the parameter input $u_t$ to encode differences among trajectories. While this design simplifies training and stabilizes the latent dynamics, it implicitly assumes that all variation in the initial conditions can be captured through the parameter space. In practice, however, many physical systems exhibit diverse and high-dimensional initial states that cannot be fully represented by a few parameters. As a result, this assumption may limit LD-EnSF’s generalization to systems with unseen or highly variable initial conditions.

In addition, the paper’s explanation of this mechanism is somewhat unclear. The statement that the initialization “flexibly accommodates varying initial conditions” is misleading, since all trajectories start from the same latent point. Figure 1 also omits how the initial latent state is introduced or how it connects to the LSTM encoder, which may confuse readers about how the model handles initial-state diversity. Clarifying this design choice—possibly with an explicit schematic or additional discussion—and considering a learned initial encoder (e.g., mapping initial full states to latent $s_{-1}$) would both improve clarity and broaden the applicability of LD-EnSF to more general dynamical regimes.

Evaluation Scope:
A minor weakness is that the experimental comparison could be broadened or analyzed further. The authors did not compare against particle filtering or variational methods (e.g., 4D-Var). It’s understandable given those struggle in such high-dimensional sparse settings, but discussing their expected performance or including them in a smaller-scale experiment would strengthen the positioning. Additionally, an ablation study on each novel component of LD-EnSF (for instance, using a static VAE-based encoder instead of LSTM, or using the ground truth dynamics vs. LDNet during assimilation) would isolate the contributions of each part. While the paper does compare LDNet vs VAE offline, it doesn’t explicitly show the effect of the LSTM encoder in the online phase versus a baseline. Such experiments (perhaps in an appendix) would provide deeper insight into how much each innovation (smooth latent dynamics, history encoder) contributes to the final performance.

Minor Presentation Issues:
There are a few presentation details that could be improved. For example, the text’s statement of speedups (e.g. “$2\times10^3$ times speedup”) seems slightly inconsistent with Table 2 values – clarifying these calculations would avoid confusion. Also, the results discussion could more explicitly highlight why LD-EnSF outperforms baselines (e.g., pointing out in text that EnSF fails due to vanishing likelihood gradients in unobserved dimensions, which LD-EnSF overcomes by using informative latent gradients). The figures are generally clear; still, adding a bit more explanation in the captions or main text for Figure 4 (e.g., explaining the behavior of “Latent-EnSF-dyn” curves, or noting that LETKF diverged in the hardest case) would be helpful. These are minor issues and easily addressable.

**Questions:**

Generality to Out-of-Distribution Dynamics: How well would LD-EnSF handle scenarios where the true dynamics deviate from the training data? For instance, if the system experiences an unforeseen regime or a parameter outside the trained range, would the assimilation degrade gracefully, or could it diverge? Did you observe any cases where the learned LDNet struggled when the truth lay outside its training distribution?

Ablation on Observation Encoder: Have you evaluated the impact of the LSTM-based observation encoder versus a simpler or non-temporal encoder? For example, how would Latent-EnSF perform if augmented with your LDNet but still using the original VAE observation encoding at each time step (ignoring history)? This would isolate the benefit of the history-aware LSTM. Similarly, what happens if observations are on a fixed grid – does the LSTM still offer advantages over a time-independent encoder?

Parameter Estimation Performance: The method jointly assimilates state and parameters. Can you provide more insight into how accurately the uncertain parameters (Re, initial bump location, forcing amplitude/spread) were estimated in your experiments? It would be useful to know, for example, the final parameter RMSE or if the filter consistently converges to near-true parameter values. This would demonstrate the effectiveness of treating $(s_t, u_t)$ together in the state vector.

Ensemble Size and Filter Stability: You mentioned that increasing ensemble size beyond a point had minimal impact on LD-EnSF’s accuracy. Could you elaborate on this? Is LD-EnSF less sensitive to ensemble size because the score-based update effectively approximates the Bayesian posterior even with few samples? Any intuition on why, say, even 1 or 5 ensemble members can yield good results would be interesting – it’s an intriguing contrast to standard EnKF which typically benefits from larger ensembles.

Computational Overheads: Table 2 shows dramatic speedups in the online phase. Could you comment on the offline costs (training LDNet and LSTM) relative to those gains? For a fair real-world assessment, one might consider how many assimilation cycles are needed to amortize the training cost. Do you envision scenarios (like repeatedly assimilating in the same system) where the upfront training is justified by many future uses of the model?

**Details Of Ethics Concerns:**

This paper does not raise any specific ethical concerns.
The study involves numerical simulations and synthetic datasets for evaluating data assimilation methods.
No human subjects, personal data, or sensitive information are used.
The work focuses on methodological and computational contributions rather than societal or safety-critical applications.

---

> ### Author Response · Authors · 2025-11-21
> **Rebuttal Reviewer TvkW (Part 1 of 3)**
>
> We thank the reviewer for the constructive suggestions which helped us to improve our paper! Below we address each point with supporting results.
>
> **Dependence on Offline Training and Generalization Limits**
>
> The fact that the model inherits the generalization limits of data driven surrogates is a drawback primarily from using a data driven approach to data assimilation as a field, and not our framework in particular. Data driven models inherently have the advantage of learning the background prior covariance by utilizing data; traditional non-data based assimilation techniques require that these covariances be hand-tuned in accordance with expert knowledge.
>
> **Novelty and Comparison with Variational Surrogate Approaches**
>
> While LD-EnSF builds on the idea of latent-space filtering introduced in Latent-EnSF, several structural differences set the two methods apart. A key distinction from Latent-EnSF is that LD-EnSF uses a dynamics-consistent latent space learned by LDNet, whereas the VAE latent in Latent-EnSF does not obey a meaningful temporal evolution and introduces drift during filtering. Second, Latent-EnSF requires decoding the full state at every assimilation step, causing high computational cost and feedback of reconstruction noise; LD-EnSF performs the entire update in latent space and decodes only once, enabling regimes that Latent-EnSF cannot handle efficiently. Finally, LD-EnSF naturally supports parameterized dynamical systems because the latent dynamics are conditioned on $u$, whereas Latent-EnSF has no mechanism to represent parametric variations.
>
> > Explicitly discussing this relationship would clarify that LD-EnSF’s main strength lies in its practical integration of latent surrogates with score-based filtering.
>
> We actually have a newer framework where we tested the corresponding assimilation speeds for LD-EnSF versus a smoothing based approach using a LDNet surrogate, and while it provides a few advantages, assimilation speed is still faster in the LD-EnSF due to 1). not needing backward propagation of gradients during assimilation time and 2). Filtering versus smoothing. On average, LD-EnSF is about 10 times faster at assimilation time.
>
> > Positioning LD-EnSF more explicitly within this broader family of latent-space Bayesian filters—highlighting differences...in this research area.
>
> We thank the reviewer for the position suggections. We have revised the Related Work section to incorporate these suggestions, including recent latent-space Bayesian filtering approaches, and clarified how LD-EnSF is positioned within this broader family.
>
> **Computational Complexity with Ensemble Size**
>
> While LD-EnSF has a per-update cost of $O(N_e T_{diff}d_s)$, with $N_e$ ensemble members and $T_{diff}$ diffusion steps and $d_s$ is the latent state dimension. The typical value for $d_s$ is 10–50 (Appendix F.1), allowing $N_e=10-20$ to be sufficient (Appendix F.2). This is consistent with prior EnSF results, where even million-dimensional systems required only modest ensemble sizes. In addition, $T_{diff}$ is also small in practice (100 steps), and ensemble members are fully parallelizable.
>
> To further clarify the positioning relative to variational baselines, we recently implemented a smoothing-based approach using the same LDNet surrogate in follow-up work. This variant benefits from gradient-based optimization but requires backward propagation of Jacobians at each assimilation step and solves a smoothing objective rather than a filtering one. In contrast, LD-EnSF avoids all gradient backpropagation during assimilation and performs closed-form likelihood updates in latent space. In our tests, LD-EnSF achieves roughly a 10× speedup over the surrogate-based smoothing method during assimilation.
>
> **Implementational Details**
>
> Though we do not provide an open source code at the moment (will provide upon acceptance), we have pseudocodes for the assimilation steps in Algorithms 1 and 2. Meanwhile, the general framework is visualized in detail in Fig. 1. Although the pipeline contains several neural components, they do not require joint training. Each module is trained independently with its own objective. This separation keeps the optimization process simple, and we further reduce practical complexity by automating hyperparameter tuning with Bayesian optimization. There are some additional training details provided in Appendix C which includes hyperparameter choices that we have made.

---

> ### Author Response · Authors · 2025-11-21
> **Rebuttal Reviewer TvkW (Part 2 of 3)**
>
> **Limitation and Clarity of Latent-State Initialization**
>
> We have updated Fig. 1 to clarify that $s_{-1}=0$. In Phase 2 of Fig. 1, the “ground-truth’’ latent states used during LSTM training are extracted from the learned LDNet, which is difficult to visualize directly. Therefore, we added additional description in the caption and Section 3.2. Since the initial physical state used for assimilation is decoded from $s_0$, the transition from $s_{-1}=0$ through the dynamics network does provide some flexibility to accommodate different initial conditions. For example, in the tsunami experiments, we varied the initial location of the Gaussian bump and obtained distinct latent trajectories (Fig. 2).
>
> We agree that relying on parameter input to distinguish initial conditions introduces limitations. We also experimented with a learned initial-state encoder, but the performance was not satisfactory. In our setting, system parameters often provide cleaner and more reliable information than high-dimensional encoded states, and LDNet is designed specifically to utilize such parameterized variations.
>
> **Evaluation Scope**
>
> We already provide ablations for the main components of LD-EnSF in Appendix G. Specifically, Appendix G.1 analyzes the improvements from the LDNet training procedure, Appendix G.2 compares the VAE-based encoder (referred to as the CNN encoder) with the LSTM encoder used in LD-EnSF, and Appendix G.4 evaluates assimilation with and without latent-space updates. Using ground-truth dynamics would reduce the method to Latent-EnSF, and the comparison to this model is included in Fig. 4. These ablations isolate the contributions of latent dynamics and the history encoder.
>
> **Minor Presentation Issues**
>
> Thanks for the typo correction, we have fixed the number to match (4x10^3). We also added more discussions on the assimilation resutls in the caption of Fig. 4 and Line 395-397.
>
> **Generality to Out-of-Distribution Dynamics**
>
> > How well would LD-EnSF handle scenarios where the true dynamics deviate from the training data?
>
> We discuss this in Appendix E.3, where the error increases as the distribution of the test data becomes more and more divergent with the training data. However, performance is still somewhat reasonable, even with heavy deterioration.
>
> **Ablation on Observation Encoder**
>
> > Have you evaluated the impact of the LSTM-based observation encoder versus a simpler or non-temporal encoder?...Similarly, what happens if observations are on a fixed grid – does the LSTM still offer advantages over a time-independent encoder?
>
> We did evaluate the performance of the non-temporal CNN/VAE-style encoders and they introduce more noise and do not perform as well as the LSTM encoder. The comparison is shown in Fig. 14 of Appendix G.2. Our primary results use a fixed observation grid to match the setting of the time-independent encoder baseline. We also evaluated irregular spatial observations (Fig. 9) to show that LD-EnSF extends naturally to more flexible scenarios.
>
> **Parameter Estimation Performance**
>
> > Can you provide more insight into how accurately the uncertain parameters (Re, initial bump location, forcing amplitude/spread) were estimated in your experiments?
>
> See the middle columns Figs. 9 and 10 in the appendix for the corresponding errors of the parameters. Depending on the experiments, the values could vary, but in general the accuracy is high.
>
> **Ensemble Size and Filter Stability**
>
> >  You mentioned that increasing ensemble size beyond a point had minimal impact on LD-EnSF’s accuracy. Could you elaborate on this? Is LD-EnSF less sensitive to ensemble size because the score-based update effectively approximates the Bayesian posterior even with few samples? Any intuition on why, say, even 1 or 5 ensemble members can yield good results would be interesting – it’s an intriguing contrast to standard EnKF which typically benefits from larger ensembles.
>
> We agree that the weak dependence on ensemble size is an interesting difference from EnKF. In LD-EnSF the ensemble is only used to estimate the prior score in a low-dimensional latent space, while the likelihood score is computed analytically from the LSTM-encoded observations and typically dominates the update. As a result, even a small ensemble provides a sufficiently accurate Monte-Carlo estimate of the prior score, so RMSE saturate quickly. For very small N (e.g., 1–5), LD-EnSF behaves more like a stochastic MAP tracker in latent space: the LDNet–LSTM representation already captures the main posterior structure, so a single trajectory is often close to the posterior mean for our benchmarks. Larger ensembles mainly improve uncertainty quantification rather than pointwise accuracy. This contrasts with EnKF, where the ensemble must approximate a high-dimensional covariance matrix, making performance much more sensitive to ensemble size.

---

> ### Author Response · Authors · 2025-11-21
> **Rebuttal Reviewer TvkW (Part 3 of 3)**
>
> **Computational Overheads**
>
> > Could you comment on the offline costs (training LDNet and LSTM) relative to those gains?... Do you envision scenarios (like repeatedly assimilating in the same system) where the upfront training is justified by many future uses of the model?
>
> In Appendix C.2, we address the computational overhead of the offline costs and the number of assimilation cycles needed for the online gain to make up for the additional training cost.

---

### Official Review · Reviewer_cjhs · 2025-10-30

**Soundness:** 2
**Presentation:** 2
**Contribution:** 2
**Rating:** 6
**Confidence:** 4

**Summary:**

The paper proposes LD-EnSF, a score-based Bayesian filtering method that performs all assimilation steps in a low-dimensional latent space learned by an improved Latent Dynamics Network (LDNet) and coupled to a history-aware LSTM observation encoder. The method extends EnSF (ensemble score filtering) to handle severe spatiotemporal sparsity without resorting to costly full-state simulations by: (i) learning smooth latent dynamics that can be time-stepped (Euler) with parameter inputs, plus a stronger reconstruction network; (ii) mapping sparse, possibly irregular observations $𝑦_{1:𝑡}$  to latent pairs $(𝑠^𝑡, 𝑢^𝑡)$ with an LSTM; and (iii) running EnSF entirely in latent space then decoding to physical space only when needed. Overall this presents a principled way of approaching the problem. Experiments on Kolmogorov flow, tsunami (shallow water), and a forced hyperviscous rotating atmosphere show lower assimilation RMSE than EnSF, Latent-EnSF (VAE), and LETKF, with very large runtime speedups because full-space propagation is replaced by latent dynamics

**Strengths:**

Decoder-free assimilation loop: All filtering happens in latent space (state + params), avoiding per-step decoding and cutting both compute and error accumulation.

Efficiency at scale: Small latent dimension + ensemble updates → orders-of-magnitude cheaper than full-state DA; design is hardware-friendly and parallelizable.

Robustness features: Reverse-SDE damping and simple latent noise modeling make the update stable under severe sparsity/noise.

Clear training recipe: Two-stage LDNet training with well-specified schedules/hparams improves reproducibility and stability.

Extensive Evaluation : Demonstrated on Kolmogorov flow, tsunami (shallow water), and a rotating atmosphere covering increasing complexity and different observation settings.

Thorough ablations: Sensitivity to latent rank, ensemble size, observation density/cadence, noise level, and architecture choices; includes OOD initial-condition tests.


Reproducibility: Detailed setups (observation models, training schedules, metrics (see Appendix) ) and systematic reporting make results believable and repeatable.

**Weaknesses:**

Overall the paper presents a clear contribution with thorough experimentation. Following are my major concerns :

Related Work Missing :

The paper under-cites several very relevant 2024–2025 works in  that would strengthen positioning:

Neural Operators for DA and Semilinear PDEs : Fourier Neural Operator and SFNO have presented great result in PDE/wether modeling but no discussion have been provided in regard to them. Additionally Semilinear Neural Operator (ICLR 2024) that proposes a recursive neural-operator framework that explicitly addresses prediction and data assimilation for semilinear PDEs have also not been cited ;
Neural Koopman priors and Koopman-based DA : Frion et al., “Neural Koopman Prior for Data Assimilation” formulates DA with a neural Koopman prior (now a TSP 2024 article), and KODA (arXiv 2024) integrates forecasting with an online data-assimilation loop using Koopman-guided components. Both connect directly to low-dimensional latent linearizable dynamics for DA, much like LD-EnSF’s latent evolution;  Modern 4D-Var and deep 4D-Var variants : The paper cites classical 3D/4D-Var (e.g., Rabier & Liu 2003) but misses recent learned or hybrid 4D-Var systems that tackle cost and sparsity with neural parameterizations—e.g., 4DVarNet (end-to-end DA backboned on variational objectives), En4DVarNet for uncertainty, 4DVarFormer (attention-based 4D-Var surrogate with rapid multivariate analyses), and operational-scale hybrids like FuXi-En4DVar. These are important baselines or at least conceptual references for the atmosphere case and the efficiency narrative.

Action item : by adding a paragraph in Related Work discussing (SNO/NO-DA, ClimODE, Koopman-DA, deep/hybrid 4D-Var), explaining what LD-EnSF gains with a training-free score component and learned latent dynamics, and why that’s preferable under extreme sparsity.

Technical Weaknesses :

Latent observation model = identity. The filter assumes $𝐻_{latent}(𝜅_𝑡)=𝜅_𝑡$, so the LSTM’s outputs $(𝑠_𝑡,𝑢_𝑡)$	​
 are taken as direct noisy measurements of the true latent pair $(𝑠_𝑡,𝑢_𝑡)$. Any encoder bias/miscalibration directly contaminates the update; there’s no learned/structured latent observation operator to absorb mismatch.

Decoder-free assimilation hides reconstruction bias. Assimilation runs entirely in latent space (a strength), but it also means reconstruction errors don’t get corrected during the loop. If the decoder has bias, final physical-space fields can drift even when latent RMSE falls. (The paper itself emphasizes that decoding is only needed at the end, which is fast, but leaves this feedback gap.)

Noise handling is crude in latent space. The latent observation noise 𝛾^_{𝑡} is estimated post-hoc and then treated as uniform across latent dimensions. That’s convenient but brittle when different latent directions have very different uncertainty.

**Questions:**

Fourier Neural Operators have shown great success recently in weather and PDE modelling. Why have no discussion been provided in that regards?

Latent noise modeling: In Eq. (8) the identity $H_{latent}$ and a scalar 𝛾^_{𝑡}  are assumed via empirical estimation. How sensitive is EnSF’s update to misspecifying 𝛾^_{𝑡} across latent dimensions? Could you learn a diagonal or low-rank covariance in latent space cost-effectively?

Are proposals for $u_{t}$ purely resampling from the previous posterior, or is there diffusion/jitter? What happens when u varies slowly or remains static?

Physical time vs latent  Δt: Since Δt is tuned, how does the method behave if the observation sampling rate changes (e.g., sparser/faster streams) without re-training?

When LDNet struggles (very long horizons, regime changes), could LD-EnSF seamlessly “fallback” to occasional full-model nudging (a la hybrid LD-EnSF/4D-Var) while keeping most steps latent?

---

> ### Author Response · Authors · 2025-11-21
> **Rebuttal Reviewer cjhs (Part 1 of 2)**
>
> We thank the reviewer for the constructive suggestions which helped us to improve our paper! Below we address each point with supporting results.
>
> **Related Work**
>
> We thank the reviewer for pointing out these important recent developments. We have updated the Related Work section to include Neural Operator–based models (FNO, SFNO, Semilinear Neural Operators), Koopman-based DA methods (Koopman Prior, KODA), and modern deep/hybrid 4D-Var systems (4DVarNet, En4DVarNet, 4DVarFormer, FuXi-4DVar). These approaches focus on learning high-dimensional surrogate dynamics or full-state variational updates. In contrast, LD-EnSF compresses the full high-dimensional field into a smooth and strongly correlated latent space. This enables information to propagate through the entire state despite limited data. Therefore, even very sparse observations can update global latent modes rather than only local grid values.
>
> **Latent Observation Model = Identity**
>
> The reviewer mentions that the latent observation function being the identity function may make the model less robust. These may introduce somewhat biased latent representations, but effects are minimal. In Appendix E.2, we conducted different noise distribution tests for the observations, where we tested the model’s robustness in the actual space to noise of the standard gaussian distribution, a nonstationary gaussian distribution, heteroskedasic noise, multivariate gaussian noise, and a beta(2, 5) noise. Though they somewhat reduced the performance, the impact is not particularly significant.
>
> **Decoder Reconstruction Bias**
>
> We agree that decoder bias is a valid concern. If the true full state were available during assimilation, a decoder-only framework would indeed be unable to use it to correct the latent state. In our setting, no full observations of states are provided during assimilation. Operating entirely in latent space in this scenario has a practical advantage compared with the decoder-encoder architecture such as Latent-EnSF. In LD-EnSF, biases in the decoder or encoder do not feed back into the update loop, so they cannot accumulate over multiple propagation steps.
>
> **Noise handling in Latent Space**
>
> > How sensitive is EnSF’s update to misspecifying $\gamma^{𝑡}$ across latent dimensions? Could you learn a diagonal or low-rank covariance in latent space cost-effectively?
>
> We added a new section (Appendix F.5, page 25) to evaluate the sensitivity of LD-EnSF with misspecified $\hat{\gamma}_t$. As shown in Fig. 12, using a $\hat{\gamma}_t$ that is smaller or larger than the estimated value causes only mild changes in the transient error, and the assimilation still converges to a similar low-error regime. This indicates that LD-EnSF is not highly sensitive to moderate misspecification of the latent noise level. Regarding learning a diagonal or low-rank covariance, this is in principle possible, but the scalar $\hat{\gamma}_t$ already provides robust performance in our experiments, and the additional parameters would meaningfully increase estimation complexity.
>
> **Time Varying $u_t$**
>
> > Are proposals for $u_t$ purely resampling from the previous posterior, or is there diffusion/jitter? What happens when u varies slowly or remains static?
>
> In this work, we are using a static $u(t)$ for the experiments. However, we have performed preliminary tests in ongoing follow-up work on systems with time-dependent forcing terms, such as precipitation inputs in flooding. In these cases, we set a digital elevation model given by an area in Texas which is prone to floods. The LDNet is able to model corresponding water height given some temporally-varying rainfall rate (defined to be constant across the spatial domain) which we define to be the forcing $u(t)$. This means that when the external forcing varies but its patterns remain within the distribution represented during training, no retraining is needed. See https://imgur.com/a/ld-ensf-rebuttal-iW9TU7A for the corresponding hydrograph at a measurement point. For use within LD-EnSF, slowly varying $u_t$ can be propagated without modification. If $u_t$ changes more drastically, then an explicit dynamical model $u_{t+1} = F_u(u_t)$ would need to be learned in addition to the latent state dynamics. We have added this clarification to the revised text in Section 4.1.

---

> > ### Author Response · Authors · 2025-11-21
> > **Rebuttal Reviewer cjhs (Part 2 of 2)**
> >
> > **Physical Time vs Latent $\Delta t$**
> >
> > > Since $\Delta t$ is tuned, how does the method behave if the observation sampling rate changes (e.g., sparser/faster streams) without re-training?
> >
> > LD-EnSF uses a learned latent dynamics model with a tunable latent time step $\Delta t$. The pretrained LDNet can be evaluated at arbitrary fractional time increments, since latent states can be propagated using any chosen $\Delta t$ (e.g. to get the latent state at time step $5.4\Delta t$ one can simply get the latent state at $5 \Delta t$ and then propagate it again by $0.4\Delta t$). Thus, changes in the physical observation rate do not require retraining LDNet. From here, only a new LSTM encoder need be adjusted to match the new sampling interval, which is very lightweight due to the low dimensionality of the sparse observations and the latent states.
> >
> > **Hybrid LD-EnSF with Full Space DA Methods**
> >
> > > When LDNet struggles (very long horizons, regime changes), could LD-EnSF seamlessly “fallback” to occasional full-model nudging (a la hybrid LD-EnSF/4D-Var) while keeping most steps latent?
> >
> > A hybrid model seems difficult to implement as there is no subsequent mapping from the full state back to the latent state. However, if the LD-EnSF fails after a certain time step, one could simply reconstruct the corresponding full state at that time step to apply a standard full-dimensional data assimilation approach.

---

### Official Review · Reviewer_ogtz · 2025-11-01

**Soundness:** 4
**Presentation:** 4
**Contribution:** 3
**Rating:** 8
**Confidence:** 3

**Summary:**

This paper proposes LD-EnSF, a score-based data assimilation framework that combines Latent Dynamics Networks (LDNets) and an LSTM-based observation encoder to perform Bayesian filtering directly in a low-dimensional latent space.

Traditional data assimilation methods such as EnKF or EnSF are computationally expensive because they operate in the full physical space and require repeated forward model simulations. LD-EnSF negates this by learning surrogate dynamics in latent space and performing all filtering steps there, using an ensemble score filter (EnSF) to update the posterior distribution. The LSTM encoder processes sparse and irregular observations and aligns them with the latent states and parameters.

Empirical validation is conducted on three different systems (Kolmogorov flow, tsunami modeling, and atmospheric dynamics) and under severe spatial and temporal sparsity. The method demonstrates strong improvements in accuracy and speed, with orders of magnitude reductions in runtime while maintaining or improving assimilation accuracy.

**Strengths:**

Clear Motivation and Relevance
- The paper tackles a key limitation of recent score-based filters, their high computational cost and poor performance with sparse observations.

Solid Technical Design
- The integration of latent surrogate dynamics (LDNet) and score-based Bayesian filtering (EnSF) is smart.
- The introduction of a history-aware LSTM observation encoder effectively extends the latent assimilation framework to handle irregular and sparse data.

Comprehensive Experiments
- The authors test across multiple physical systems of increasing complexity.
- Results include both structured and unstructured observation setups, multiple levels and differing types of noise, and sensitivity analyses.

Extensive Ablation and Robustness Studies
- Appendices systematically evaluate noise robustness, out-of-distribution generalization, latent dimension sensitivity, ensemble size, and architectural design choices.

**Weaknesses:**

Limited Theoretical Novelty
- The proposed method primarily combines existing techniques (LDNet, EnSF, LSTM encoding). While the combination is well-executed and impactful the theoretical advancement is modest. The novelty lies more in the integration and empirical rigor.

Benchmark Coverage and Positioning
- Comparisons are limited to EnSF, Latent-EnSF, and LETKF. While these are strong and relevant baselines, the paper could benefit from a clearer discussion of recent efficient variational and diffusion-based data assimilation approaches, such as Tensor-Var: Efficient Four-Dimensional Variational Data Assimilation (Yang et al., 2025) and DiffDA: A Diffusion Model for Weather-Scale Data Assimilation (Huang et al., 2024). Although implementing these methods in the current setup may not be straightforward, a more explicit positioning in the Related Work section would help situate LD-EnSF within the broader landscape.

**Questions:**

1. Theoretical Guarantees:
Can the authors comment on whether any convergence guarantees exist for the latent-space assimilation process, particularly as a function of the surrogate model error?

2. Dynamic Re-training:
How feasible is LD-EnSF in systems where governing dynamics change over time? Could partial retraining or online fine-tuning mitigate the offline cost?

3. Uncertainty Quantification:
How reliable are uncertainty estimates when mapped back to physical space?

---

> ### Author Response · Authors · 2025-11-21
>
> We thank the reviewer for the constructive suggestions which helped us to improve our paper! Below we address each point with supporting results.
>
> **Benchmark Coverage and Positioning**
>
> Tensor Var is also a smoothing algorithm as compared to filtering algorithm, but we have added a section on Tensor-Var in the introduction and related work. We alse expanded upon the DiffDA discussion in the related work section.
>
> **Theoretical Guarantees**
> > Can the authors comment on whether any convergence guarantees exist for the latent-space assimilation process, particularly as a function of the surrogate model error?
>
> To our knowledge, a full convergence theory for latent space data assimilation with learned neural surrogates is till largely open. Existing works provide only partial guarantees. For example, Cheng et al. [1] derives an upper bound on the difference between the surrogate and original latent-space assimilation cost functions. Recent method such as Mücke et al. [2] and Si et al. [3] focus on empirical convergence and scalability. Li et al [4] provide theoretical guarantees only for the surrogate representation, rather than for the latent assimilation procedure itself. While we are unable to give a full convergence analysis of LD-EnSF, we include several empirical assessments in Appendix F, including sensitivity to latent dimension, ensemble size, surrogate model, and observation density.
>
> **Complex Dynamics and Re-training**
> >How feasible is LD-EnSF in systems where governing dynamics change over time? Could partial retraining or online fine-tuning mitigate the offline cost?
>
> Our current experiments focus on systems with fixed governing equations and fixed parameters. In addition, we have performed preliminary tests in ongoing follow-up work on systems with time-dependent forcing terms, such as precipitation inputs in flooding. In these cases, we set a digital elevation model given by an area in Texas which is prone to floods. The LDNet is able to model corresponding water height given some temporally-varying rainfall rate (defined to be constant across the spatial domain) which we define to be the forcing u(t). This means that when the external forcing varies but its patterns remain within the distribution represented during training, no retraining is needed. See https://imgur.com/a/ld-ensf-rebuttal-iW9TU7A for the corresponding hydrograph at a measurement point.
>
> When the governing equations themselves change over time (for instance, bathymetry changes due to erosion in the flooding case), all surrogate-based data assimilation methods face clear difficulty unless some adaptation is introduced. LD-EnSF offers several practical adaptation paths. The LDNet can be updated by training only selected components, for example by freezing the reconstruction network or the dynamics network and updating the other. Because the LDNet structure can also be easily blend together with physics-informed losses, it can also be retrained using partial data when full data sets are not available. After updating the LDNet, retraining the LSTM observation encoder is straightforward and has a low computational cost.
>
> **Uncertainty Quantification**
> > How reliable are uncertainty estimates when mapped back to physical space?
>
> Whether the latent uncertainty can be distorted by the decoder mapping is a natural concern. In Fig. 9 of Appendix E.3, we have the corresponding error bars with respect to both the latent state (right) and after decoding to the original space (left). One can see that the two are strongly correlated: ensemble members with smaller error bars in the latent space beget a corresponding amount of error in the physical space. This shows that the decoder exhibits a smooth behavior and does not mutate the uncertainty in unexpected manners. Moreover, the reconstruction errors of the reconstruction network are substantially smaller than the assimilation errors, so the dominant source of uncertainty stems from the data assimilation process rather than the latent to physical mapping.
>
> Reference
>
> [1] Cheng, S., Chen, J., Anastasiou, C., Angeli, P., Matar, O. K., Guo, Y.-K., Pain, C. C. & Arcucci, R. (2023). Generalised Latent Assimilation in Heterogeneous Reduced Spaces with Machine Learning Surrogate Models. Journal of Scientific Computing, 94: 11. https://doi.org/10.1007/s10915-022-02059-4
>
> [2] Mücke, Nikolaj T., Sander M. Bohté, and Cornelis W. Oosterlee. "The deep latent space particle filter for real-time data assimilation with uncertainty quantification." Scientific Reports 14, no. 1 (2024): 19447.
>
> [3] Si, Phillip, and Peng Chen. "Latent-EnSF: A Latent Ensemble Score Filter for High-Dimensional Data Assimilation with Sparse Observation Data." The Thirteenth International Conference on Learning Representations (2025).
>
> [4] Li, Zhuoyuan, Bin Dong, and Pingwen Zhang. "Latent assimilation with implicit neural representations for unknown dynamics." Journal of Computational Physics 506 (2024): 112953.

---

### Meta-Review · Area_Chair_F832 · 2026-01-05

**Summary:**

The reviewers acknowledged solid experimental work and significant computational speedups but raised concerns about novelty and theoretical justification. Reviewer ogtz (score 8) found the integration well-executed with comprehensive experiments but noted limited theoretical novelty. Reviewer cjhs (score 6) appreciated the extensive evaluation but identified missing citations to recent work in neural operators, Koopman-based DA, and modern 4D-Var variants, plus technical concerns about crude latent noise handling. Reviewer TvkW (score 4) questioned incremental novelty over Latent-EnSF, noted dependence on offline training with unclear generalization limits, and requested more explicit computational complexity analysis. Reviewer YcmD (score 2) argued the LSTM's time-delay embedding, not the latent representation itself, addresses sparsity, and requested clarification on claims about "ill-posed scores" and "informative gradients."

**Reviewer Concerns:**

The rebuttal addressed several concerns. For YcmD, clarifications about vanishing gradients and the addition of Figure 13 comparing fixed versus trainable Fourier features resolved major issues, with the reviewer stating "I have accordingly increased my scores." For ogtz, who requested comparison with Tensor-Var, and cjhs, who asked for 4DEnVar results, the authors added these discussions and experiments. The authors also expanded the related work section to include neural operator and Koopman-based methods as requested by cjhs, and added sensitivity tests for latent noise misspecification in Appendix F.5.

However, concerns about incremental methodological novelty persist, with multiple reviewers noting the work primarily combines existing techniques. TvkW's questions about when computational advantages exceed variational baselines using similar surrogates and the unclear latent initialization mechanism were only partially addressed.

**Reviewer Scores:**

Reviewer ogtz would maintain their score of 8.

Reviewer cjhs would either maintain their score or increase from 6 to 8 given the expanded comparisons and related work.

Reviewer TvkW would likely increase from 4 to 6, acknowledging improved clarity.

Reviewer YcmD indicated updating from 2 to 6.

Hypothetical average score: 5-6.5. Recommendation: accept. The paper represents a successful application of machine learning to data assimilation.

---

### Decision · Program_Chairs · 2026-01-26

Accept (Poster)